# Don't Waste Your Time: Early Stopping Cross-Validation

**Edward Bergman**[1, *]  **Lennart Purucker**[1, *]  **Frank Hutter**[2, 1]

[1]University of Freiburg
[2]ELLIS Institute Tübingen

**Abstract**  State-of-the-art automated machine learning systems for tabular data often employ cross-validation; ensuring that measured performances generalize to unseen data, or that subsequent ensembling does not overfit. However, using k-fold cross-validation instead of holdout validation drastically increases the computational cost of validating a single configuration. While ensuring better generalization and, by extension, better performance, the additional cost is often prohibitive for effective model selection within a time budget. We aim to make model selection with cross-validation more effective. Therefore, we study early stopping the process of cross-validation during model selection. We investigate the impact of early stopping on random search for two algorithms, MLP and random forest, across 36 classification datasets. We further analyze the impact of the number of folds by considering 3-, 5-, and 10-folds. In addition, we investigate the impact of early stopping with Bayesian optimization instead of random search and also repeated cross-validation. Our exploratory study shows that even a simple-to-understand and easy-to-implement method consistently allows model selection to converge faster; in ~94% of all datasets, on average by 214%. Moreover, stopping cross-validation enables model selection to explore the search space more exhaustively by considering +167% configurations on average within one hour, while also obtaining better overall performance.

## 1 Introduction

Automated machine learning (AutoML) systems search for the optimal machine learning model for a given task. Therefore, AutoML systems must validate the performance of a potential hyperparameter configuration by employing methods such as holdout or cross-validation. State-of-the-art AutoML systems for tabular data often employ cross-validation; ensuring that measured performances generalize to unseen data (Vakhrushev et al., 2021; Gijsbers and Vanschoren, 2021; Feurer et al., 2022), or that subsequent ensembling does not overfit (Erickson et al., 2020).

The disadvantage of using $k$-fold cross-validation instead of holdout validation is a drastic increase in the computational cost of validating a single configuration. It requires training and evaluating $k$ models instead of only one. Moreover, cross-validation is often used with many folds (e.g., $k = 8$ in AutoGluon (Erickson et al., 2020)); incurring an additional $O(N)$ cost, scaling linearly with the fold count. To illustrate, fitting and validating the configuration of an MLP was empirically ~10.5× more expensive when going from a 90/10 holdout validation to 10-fold cross-validation on the *okcupid-stem* dataset with 20 features and 50789 data points.

While ensuring better generalization and, by extension, better performance, the additional cost is often prohibitive for effective model selection within a time budget. Better-performing configurations might not be evaluated within the time budget, as the additional cost prohibits an exhaustive exploration of the search space. Similarly, model selection takes longer to converge on validation data, as the time needed to reach convergence increases when using cross-validation. In effect, this implies that a $k×$ increase in time is needed for $k$-foldcross-validation to evaluate the same number of configurations as holdout validation during a random search.

---

*Equal contribution.

We aim to make model selection with cross-validation more effective for AutoML. In this exploratory study, we investigate *early stopping the process of cross-validation* during model selection. We hypothesize that early stopping enables model selection to explore the search space more extensively and converge faster. Moreover, we hypothesize, that even simple-to-understand and easy-to-implement methods can achieve both.

Our study is motivated by the straightforward observation that many configurations evaluated during a random search or Bayesian optimization (BO) are worse than the incumbent configuration (the best one seen so far). Consequently, many configurations are expensively evaluated but discarded. Ideally, early stopping $k$-fold cross-validation, e.g., after the first fold, could allow us to discard configurations without the unnecessary overhead of validating on $k - 1$ folds.

In addition, we were motivated by the current landscape of AutoML systems. One of the first popular AutoML systems was *in 2013*, Auto-Weka (Thornton et al., 2013), which used racing, a method to early stop validation of hyperparameter configurations. Yet, *none* of the state-of-the-art AutoML systems compared as part of the AutoML benchmark (Gijsbers et al., 2022) use any method for early stopping of cross-validation. Thus, we saw the need to revisit this concept in 2024.

Finally, we were motivated to focus on simple-to-understand and easy-to-implement methods to improve the meaningfulness of our exploratory study. Such methods have minimal confounding factors, unlike methods with hyper-hyper-parameters or multiple different options for their implementation. Moreover, easy-to-implement methods allow for quicker adaption by existing state-of-the-art AutoML systems for tabular data.

We revisit the concept of early stopping cross-validation during model selection in our exploratory study by investigating its effect on random search. Therefore, we evaluate two simple-to-understand and easy-to-implement methods for early stopping cross-validation and compare them with the traditional approach of not using early stopping during model selection. We compare the methods for two algorithms, MLP and random forest (RF), across 36 classification datasets from the AutoML Benchmark (Gijsbers et al., 2022). We further analyze the impact of the number of folds by considering 3-, 5-, and 10-fold cross-validation. In addition, we investigate the impact of early stopping on Bayesian optimization and also repeated cross-validation.

Our study shows that a simple-to-understand and easy-to-implement method for early stopping cross-validation (**1**) consistently allows model selection with random search to converge faster, in ~94% of all datasets, on average by 214%; and (**2**) explore the search space more exhaustively by considering +167% configurations on average within the time budget of one hour; while also (**3**) obtaining better overall performance.

**Our contributions**. We provide (**A**) evidence on the advantages of early stopping for cross-validation for AutoML systems in 2024 and beyond, (**B**) a simple-to-understand, easy-to-implement, and well-performing method for early stopping, and (**C**) a reproducible and extensible framework for future research on early stopping cross-validation[1].

## 2 Related Work

Several methods comparable to early stopping of cross-validation have previously been used in AutoML. Auto-WEKA (Thornton et al., 2013) relied on random online aggressive racing (ROAR) implemented in SMAC (Hutter et al., 2011) to perform *intensification*. This intensification progressively evaluates folds, referred to as *instances* in the racing literature, to quickly discard configurations that are deemed not worth investing further resources in. One type of racing method is based on statistical tests, such as *F-Race* (Birattari et al., 2002, 2010). This statistical racing approach has been combined with an iterative search for better configurations in *Iterated F-Race* (Balaprakash et al., 2007; Birattari et al., 2010), short *irace* (López-Ibáñez et al., 2011; López-Ibáñez et al., 2016).

---

[1] `https://github.com/automl/DontWasteYourTime-early-stopping`

FocusedILS (Hutter et al., 2009) found racing based on statistical tests to be too conservative. As this does not allow stopping configurations after a single fold, even if that fold's performance is so poor that the resulting upper bound on the configuration's $k$-fold cross-validation score is already lower than the incumbent's actual score (e.g., a single fold with score 0.1, with the incumbent having a 3-fold CV score of 0.8). To fix this, both FocusedILS and ROAR *aggressively* reject configurations if they are worse than the incumbent based on their evaluated $n$ folds, even if $n$ is as small as one.

More recently, Vieira et al. (2021) specifically investigated irace for automated machine learning; proposing *iSklearn* and showing competitive performance with Auto-Sklearn (Feurer et al., 2019, 2022) across eight datasets. However, iSklearn never transitioned to a supported and usable AutoML system and has not been compared to other state-of-the-art AutoML systems on a more representative benchmark, such as the AutoML benchmark (Gijsbers et al., 2022).

Although closely related, our exploratory study differs from the intensification and racing literature in two critical properties. First, several racing methods, for example, F-race, Iterated F-Race, or the irace package (López-Ibáñez et al., 2016), emphasize creating sampling distributions from which to sample future configurations. Consequently, such racing methods are incompatible with more general model selection strategies that are of interest to this study and often used in AutoML, like random search, BO, or portfolio learning (Feurer et al., 2022; Salinas and Erickson, 2023). Second, typical implementations of cross-validation in (Auto)ML rely on a worker *fully* evaluating a configuration on all folds in sequence (Feurer et al., 2019, 2022) or parallel (Erickson et al., 2020; Pedregosa et al., 2011). However, racing methods rely on being able to evaluate a model on a specific fold and resuming evaluation of that model on another fold at a later point.

As a result, adapting racing methods for model selection as used in state-of-the-art AutoML systems is not straightforward. In addition to sampling distributions, racing based on statistical tests, such as F-race, is not applicable due to commonly having to compare less than the minimum number of repeated measurements required for the tests (e.g., $n > 6$ for the Friedman test). Likewise, methods such as FocusedILS or ROAR were originally not designed to evaluate all the folds of one configuration fully. To nevertheless be able to compare to the previous work in our study, we amended ROAR to the current landscape of AutoML systems, as described in the next section.

Also related to our study are concepts from the field of multi-fidelity hyperparameter optimization (MF-HPO) (Li et al., 2017; Falkner et al., 2018; Mallik et al., 2024). Similarly to early stopping cross-validation, MF-HPO can also be used to make model selection more effective by speeding up validating configurations. When subsets of the data are treated as a fidelity, MF-HPO approaches can be adopted and also compared to racing methods. This avenue was investigated by Krueger et al. (2015) and more recently byMohr and van Rijn (2023), showing promising results and potential alternatives to traditional cross-validation. With our focus on AutoML systems, of which, to the best of our knowledge, none use alternatives to cross-validation, we refrain from including such alternatives in the scope of our study. We also refrain from including related MF-HPO methods, like Hyperband (Li et al., 2018), in our exploratory study as we specifically aim to investigate whether simple-to-understand and easy-to-implement baselines, those with minimal confounding factors and which existing AutoML systems could quickly adapt, make model selection more effective[2].

## 3 Methodology

We present the design of our exploratory study by first describing and formalizing two early stopping methods and then detailing our chosen experimental setup.

### 3.1 Early Stopping Methods

To explore the effect of early stopping cross-validation on model selection, we consider two straightforward heuristic methods, one that is more ***aggressive*** - in terms of when deciding to

---

[2]See Appendix E.1 for a supplementary discussion on why enabling adaption by existing AutoML systems motivates our work.

stop early - and one that is more ***forgiving***. In other words, we consider one method with a high threshold and another with a low threshold to determine whether a configuration should be stopped early during cross-validation. We use these two methods to drive our exploratory study by comparing them to a baseline of no early stopping, allowing us to view the behavior of model selection w.r.t. different early stopping strategies.

**Notation.** We assume the setting of $k$-fold cross-validation with $k \in \mathbb{N}_{>1}$, whereby all folds $F_{c_j} = \{f_1^{c_j}, \ldots, f_k^{c_j}\}$ are evaluated in ascending sequence for a configuration with index $j$ from an infinite search space of potential configurations $C = \{c_0, \ldots, c_{+\infty}\}$. We define the shorthand for the score (higher is better) of a configuration $c_j$ for a fold $f_i^{c_j}$ as $s^{c_j,i}$. Moreover, we assume that several different configurations might be evaluated in parallel for a time series $T \subset \mathbb{N}_{>0}$. Then, at time $t \in T$, we define the set of all fully evaluated configurations until $t$ as $C_t = \{c_j \mid c_j \in C \wedge c_j \text{ fully evaluated before } t\}$. Finally, we assume that we want to check whether cross-validation should be stopped for the first time after evaluating the first fold. Therefore, we define an early stopping method for $k$-fold cross-validation, the current fold $n$ ($1 \leq n < k$), and the configuration $c_j$ as a function $E : \{C_t, \{s^{c_j,i} \mid i \leq n\}\} \rightarrow \{true, false\}$, where $true$ tells us to stop cross-validation. To summarize, our methods rely only on a collection of previously evaluated configurations $C_t$, the scores of the current configuration up to the last evaluated fold $\{s^{c_j,i} \mid i \leq n\}$, and an early stopping function $E$.

We further introduce the mean score of a configuration $c_j$ after being evaluated on $n$ folds as: mean-score$_n^{c_j} = \frac{1}{n} \sum_{i=1}^n s^{c_j,i}$, and define the mean score of the incumbent configuration, the best-observed configuration so far, at time point $t$ as: mean-score$_t^* \in \max_{c_j \in C_t}$ mean-score$_k^{c_j}$.

**Aggressive Early Stopping.** The aggressive early stopping method stops cross-validation of a configuration, if the mean score of all already evaluated folds is worse than or equal to the mean score of the incumbent configuration. Formally, after evaluating $n$ folds and at time point $t$, we define $E_{aggressive}$ by:

$$E_{aggressive}(C_t, \{s^{c_j,i} \mid i \leq n\}) = \begin{cases} true, & \text{if mean-score}_n^{c_j} \leq \text{mean-score}_t^* \\ false, & \text{otherwise,} \end{cases} \tag{1}$$

which follows the criterion used in FocusedILS and ROAR. There are minor differences in other parts of the algorithm; for example, we always evaluate all folds for the incumbent.

**Forgiving Early Stopping.** The forgiving early stopping method stops cross-validation of a configuration, if the mean score of all already evaluated folds is worse than or equal to the worst individual fold score obtained by the incumbent configuration. Formally, similar to the above, we define $E_{forgiving}$ as:

$$E_{forgiving}(C_t, \{s^{c_j,i} \mid i \leq n\}) = \begin{cases} true, & \text{if mean-score}_n^{c_j} \leq \text{worst-score}_t^* \\ false, & \text{otherwise;} \end{cases} \tag{2}$$

whereby worst-score$_t^* = \min \{s^{c_*,1}, \ldots, s^{c_*,k}\}$ with $c_* \in \arg\max_{c_j \in C_t}$ mean-score$_k^{c_j}$.

Both of these are meant as simplistic baselines, and we nevertheless demonstrate that they can lead to substantial improvements. We expect our choices of thresholds to lead to different behaviors, with either an aggressive or forgiving phenotype. By design, we assume that $E_{aggressive}$ stops and discards more configurations than $E_{forgiving}$, due to the higher threshold to beat. Consequently, $E_{aggressive}$ should also begin to validate more configurations. This comes with the comparative downside that this threshold may be too high, especially in scenarios where the variance in fold scores is high. In contrast, $E_{forgiving}$ should be much less aggressive but incur more cost due to potentially stopping and discarding configurations later than $E_{aggressive}$.

In the context of the methodology defined by Hutter et al. (2009) for ParamILS, $E_{aggressive}$ and $E_{forgiving}$ are comparative functions that are used to determine whether one configuration is better than another. $E_{aggressive}$ and $E_{forgiving}$ compare different aspects of the performance of the incumbent configuration with the configuration currently cross-validated. The notable difference from the racing literature is that here, we allow configurations to be compared across different instance amounts: We use k folds/instances of the incumbent configuration to compute the threshold, while we use only $n$ folds/instances of the currently cross-validated configuration.

The primary intuition for why these straightforward early stopping methods may perform better than no early stopping is that the average over $n$ fold scores of a configuration might often be representative of the average over its $k$ fold scores when compared to the incumbent configuration's $k$ fold scores. Thus, we can compare the $n$-fold average of the currently cross-validated configuration with the $k$-fold average of the incumbent configuration. This is related to the insights by Mallik et al. (2024) for PriorBand (an extension of Hyperband (Li et al., 2018)), which illustrate that the correlation of the ranking of configurations in lower fidelities with the ranking in higher fidelities is indicative of how much better Hyperband is than random search (see Figures 9-11 in the appendix of PriorBand). Thereby, a lower correlation indicates worse performance for Hyperband. In a similar vein, our intuition assumes that the ranking when mixing fidelities, i.e., the currently evaluated configuration's n-fold average vs. the incumbent's k-fold average, is representative of the ranking with the highest fidelity, i.e., k-fold vs. k-fold averages. We show an empirical demonstration of the assumption behind this intuition in Appendix D.1.

### 3.2 Experimental Setup

To explore the effect of early stopping cross-validation on model selection for AutoML, we require a combination of the following components for our experiments: **A)** model selection strategies, **B)** search spaces, **C)** cross-validation scenarios, and **D)** datasets and evaluation.

**A) Model Selection Strategies**. Our experiments focus mainly on using random search as a model selection strategy. We chose random search to quantify the effect without the bias that Bayesian optimization (BO) might introduce into model selection. Nevertheless, we also investigate the effect of early stopping cross-validation on BO in an additional study.

A design choice that must be considered when utilizing BO, is what to report as the score to the BO method itself after early stopping and discarding a configuration. When using early stopping with BO, the reported performance measurements of fully evaluated and early stopped configurations are not equally informative, a typical assumption held by BO methods. We later compare two approaches: report the configuration as failed, in which case a typical approach is to report the worst possible score, or to report the mean of the folds that have been successfully evaluated (i.e., mean-score$_n^{c_j}$). We use SMAC3 (Lindauer et al., 2022) for BO.

We give all model selection strategies a time budget of one hour, a memory budget of 20GB, and four CPU cores. We perform all our experiments on Intel(R) Xeon(R) Gold 6242 CPUs @ 2.80GHz.

**B) Search Spaces**. We perform model selection, specifically hyperparameter optimization (HPO) (Feurer and Hutter, 2019), with two separate search spaces, one of which is an MLP from scikit-learn (Pedregosa et al., 2011) and the other a random forest (RF) (Breiman, 2001). Each search space also includes basic feature preprocessing. For full details on search spaces, see Appendix A.2.

We do not focus on combined algorithm selection and hyperparameter optimization (CASH, Thornton et al. (2013)) in this study and leave it for future work. We believe that it is important to first show the feasibility of early stopping cross-validation for HPO before investigating CASH because CASH introduces additional hurdles, which could be confounders. E.g., for CASH we might need to overcome algorithm-wise heteroscedastic noise in validation scores.

**C) Cross-Validation Scenarios**. We investigate three different cross-validation scenarios (for *inner* cross-validation, i.e., the cross-validation used during model selection, and not *outer* cross-validation, i.e., the cross-validation used to estimate the performance of model selection on a dataset). As early stopping saves more time with more folds, we select a small, medium, and large number of folds: 3, 5, and 10 folds. In addition, we investigate the impact of repetitions during cross-validation with 2-repeated 5-fold and 2-repeated 10-fold cross-validation. We always use stratified splitting (Appendix A.3), as is common practice (Pedregosa et al., 2011; Erickson et al., 2020)

**D) Datasets and Evaluation**. We evaluate our early stopping methods on 36 classification datasets selected from the AutoML Benchmark (Gijsbers et al., 2019) using 10-fold *outer* cross-validation. The split into a training set and a test set per fold of the outer cross-validation are provided by OpenML (Vanschoren et al., 2014). We then split the training set per outer fold into our respective *inner* cross-validation scenarios when performing model selection. We use the predictions on the *outer* test set to estimate the generalization performance of `Aggressive`, `Forgiving`, and no early stopping.

We selected the datasets that we believed to be sufficiently explored by model selection within our resource constraints, so that our results are not affected by the noise of underexplored search spaces and model selection that is too far from convergence. Therefore, we selected datasets for which random search could evaluate at least 50 configurations on average within a predefined resource budget; see Appendix A.1 for details.

We use ROC AUC to measure performance with one-vs-rest for multiclass. We normalize scores per dataset before averaging across datasets; see Appendix A.5 for details. Unless otherwise specified, reported measurements are the average across all 10 outer folds of all 36 datasets.

In the following result section, we focus on the validation scores and only later report test scores to avoid a major confounding factor when studying early stopping for cross-validation. We aim to explore whether early stopping makes model selection more effective (i.e., better at solving the inner optimization problem) within a time budget. However, we do not aim to explore whether early stopping improves or worsens the generalization of model selection.

## 4 Results

We present our main results in the following order: first, the speed of convergence; second, the exploration of the search spaces; third, the overall performance. Subsequently, we briefly summarize the results of the use of Bayesian optimization and the impact of repeating *inner* cross-validation. From now on, for ease of reading, we also refer to model selection without early stopping for cross-validation as the *baseline* and abbreviate no early stopping with `No ES` when needed.

**Hypothesis 1**. *Early stopping converges faster to the same or a better result than not stopping early.*

Table 1 demonstrates, that, for 3-, 5- and 10-fold cross-validation, both early stopping methods lead to substantial speedups, but `Aggressive` fails to match or improve over the best performance found by `No ES` in roughly half of the datasets. `Forgiving` only fails in a few cases. While both methods differ in behaviour, they're individual behaviour is consistent between MLP and RF.

Next, in Figure 1, we present a dataset-specific view of the simple early stopping methods' speedups and failure cases for the MLP with 10 folds. We show the time at which `No ES` reached its best performance versus when each method reached the same performance. We provide more plots for other cross-validation scenarios and RF in Appendix B.1. In Figure 1, we observe that `No ES` often found better models shortly before the timeout, indicating that it has likely not converged.

Although the large speedups are not unfamiliar, as demonstrated by Birattari et al. (2002) and Hutter et al. (2009), the failure count is perhaps surprising. One intuitive explanation is that `Aggressive` early stops configurations that would otherwise yield good performance, while also finding nothing it considers better in exploring the rest of the search space more exhaustively.

Table 1: **Overview of Speedups and Failures**: Average speedups achieved by the two early stopping methods, indicating how much faster a given method managed to achieve the same or better ROC-AUC than the best obtained by the baseline `No ES` within the given time budget, as well as the count of datasets failed, which is when a given method failed to do so before the time budget elapsed. Mean and standard deviations for a method are aggregated across only those datasets which were not considered as failed for that particular method. For example, the speedup of `Aggressive` with 3 fold cross-validation on the MLP search space was ~2× faster than `No ES`, when aggregated across the 15/36 datasets in which it managed to find a configuration that matched or beat `No ES`'s best performing configuration.

| Model | $k$ | Aggressive | | Forgiving | |
|---|---|---|---|---|---|
| | | Average Speedup % | Datasets Failed | Average Speedup % | Datasets Failed |
| MLP | 3 | 192% ± 45% | 21/36 | 181% ± 57% | 6/36 |
| | 5 | 196% ± 89% | 18/36 | 194% ± 59% | 1/36 |
| | 10 | 301% ± 187% | 20/36 | 174% ± 64% | 0/36 |
| RF | 3 | 190% ± 50% | 16/36 | 204% ± 67% | 3/36 |
| | 5 | 250% ± 92% | 14/36 | 267% ± 106% | 2/36 |
| | 10 | 309% ± 171% | 12/36 | 262% ± 166% | 1/36 |
| Mean Over $k$ | | 240% | 17/26 | 214% | 2/36 |

**Hypothesis 2**. *Early stopping allows model selection to explore the search space more exhaustively.*

We present the mean number of configurations considered during model selection in Table 2. Both methods evaluate at least twice as many configurations as `No ES`. Moreover, `Aggressive` also explores much more than `Forgiving`, especially for more folds.

To complement the above, in Figure 2, we present a configuration footprint plot, using a multi-dimensional scaling (MDS) embedding into 2 dimensions (Kruskal, 1964; Groenen and van de Velden, 2016; Xu et al., 2016), where `Aggressive` fails to outperform `No ES`; we present details and more examples in Appendix B.2. `Aggressive`, while having a much higher sampling density than `No ES`, rarely fully evaluates a configuration. Conversely, `Forgiving`, which has a much lower sampling density, fully evaluates more configurations than `Aggressive`. This highlights the trade-off early stopping methods must face, that is, be aggressive enough to stop configurations early, yet ensure that they are not discounted too easily. Nevertheless, both early stopping methods allow random search to explore more exhaustive exploration, albeit once negatively (`Aggressive`) and once positively (`Forgiving`).

Table 2: **Evaluated Configurations**: The mean number of configurations with at least one fold evaluated.

| Model | $k$ | Aggressive | Forgiving | No ES |
|---|---|---|---|---|
| MLP | 3 | 7928 | 6562 | 2823 |
| | 5 | 6383 | 3704 | 1460 |
| | 10 | 5279 | 1607 | 706 |
| RF | 3 | 6588 | 5765 | 2275 |
| | 5 | 5897 | 4295 | 1274 |
| | 10 | 5121 | 2528 | 610 |
| Mean Over $k$ | | 6199 | 4076 | 1524 |
| +% to No ES | | +308% | +167% | - |

**Hypothesis 3**. *Early stopping leads to better overall performance than not stopping early.*

We have illustrated that failure may occur. However, this does not explain how much worse a failing early stopping method is relative to `No ES`. In Figure 3, we show that `Forgiving` better solves the inner optimization problem and achieves a better overall validation performance than `Aggressive` and `No ES`. In particular, `Aggressive` consistently fails to beat `No ES`. This is likely

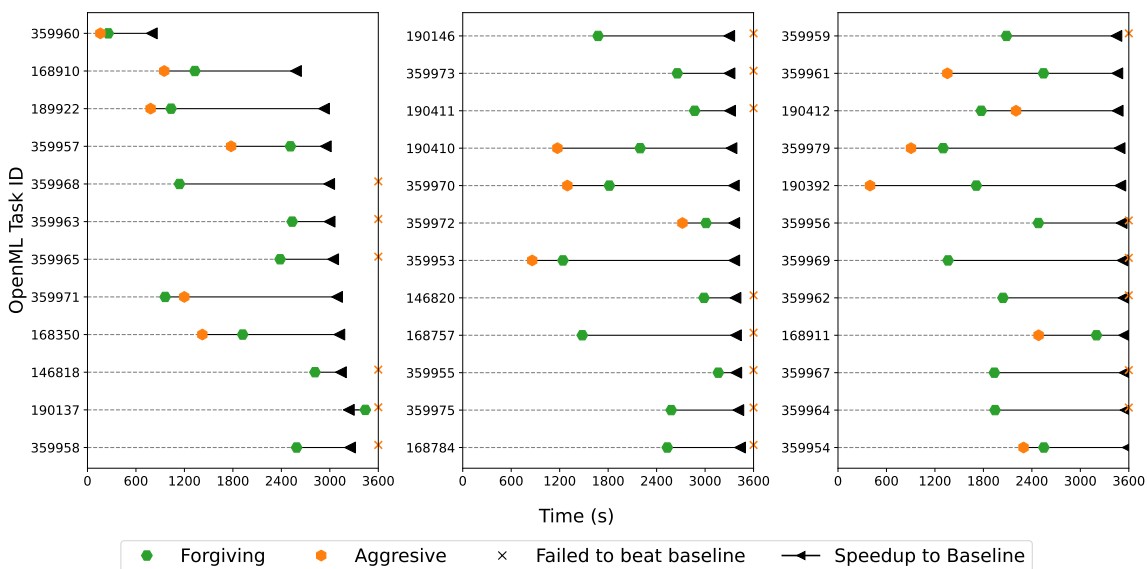

Figure 1: **Speedup Overview Per Dataset for MLP with 10 folds:** The time point of matching the best performance of `No ES` for each method per dataset. A marker indicates when a method reached the same or better performance than `No ES`. The solid black line visualizes the time saved by the fastest early stopping method per dataset.

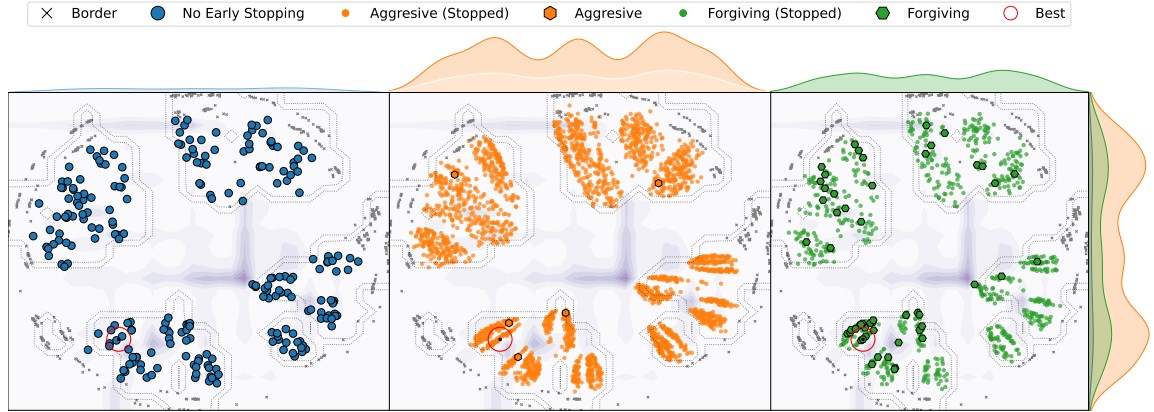

Figure 2: **Footprint Plot for OpenML Task ID 168350 on outer fold** 7: A configuration footprint, that is, a multi-dimensional scaling (MDS) embedding of the high dimensional search space for MLP to a 2-dimensional one, showing the landscape of evaluated configurations that were either evaluated (big marker) or stopped early (small marker). Darker areas represent better-performing parts of the landscape, as estimated by a Random Forest surrogate trained on the configurations with a known performance. A red circle represents the area of the landscape centered around the incumbent configuration. An x indicates a border configuration. A border configuration is sampled from the edges of the conditional search space and helps the MDS embedding to separate clusters of related configurations, i.e., those sharing related preprocessing steps. Dashed lines show the boundary of viable configurations in the 2-dimensional MDS space, as estimated by a Random Forest, trained to predict if a configuration lies within a boxed region of the space. The plot margins indicate sampling density along the given axis. The sampling density is non-uniform even when using random search due to the scaling performed by MDS. This particular footprint example is an instance where `Aggressive` failed to outperform `No ES`.

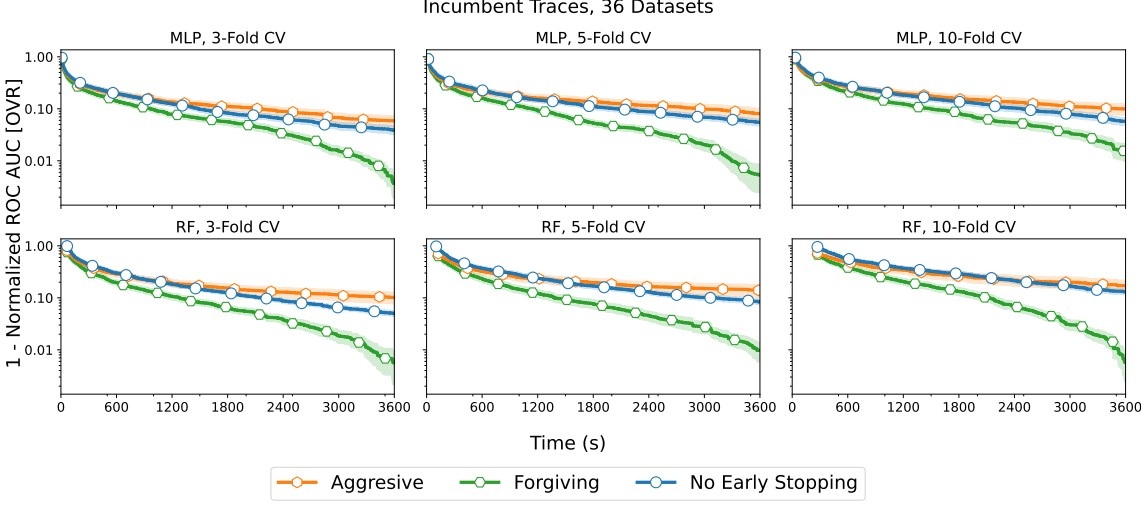

Figure 3: **Validation Performance Over Time:** The validation score incumbent trace for each method and cross-validation scenario. The normalization of ROC AUC is explained in Section 3.2

a consequence of even the best performing configuration sometimes exhibiting variation in its fold scores, enough such that `Aggressive` will early stop them before they are fully evaluated. For example, when considering the best configuration found by `No ES` for the MLP pipeline, `Aggressive` has already early stopped them on the first fold, in $40\% \pm 15\%$ of the cases.

We can observe this behaviour in greater detail with the example in Figure 8a in the appendix, where `Aggressive` evaluated only 5 configurations to completion in total; excluding the best configuration.

Furthermore, we present an example of the test, i.e., *outer* cross-validation, score compared to the validation score in Figure 4; the other cross-validation scenarios are presented in Appendix B.3. We clearly observe that improvements in validation scores do not generalize to test scores for the MLP, but do generalize for RF. We believe that the observed *algorithm-specific overfitting* confirms our reservations about the confounding effect of inspecting test scores and the need to investigate HPO first instead of CASH; as mentioned in Section 3.2,

**Additional Experiment 1.** *Early stopping cross-validation of model configurations; replacing random search with **Bayesian optimization (BO)**.*

In Appendix C.1, we present results to investigate the previous setups but utilizing BO as the model selection strategy, rather than random search. This experiment shows that applying `Forgiving` with BO can also lead to better overall performance, although to a lesser extent than for random search. With BO as the model selection strategy, `Aggressive` performs dramatically worse than `No ES`. Interestingly, we observe that reporting stopped configurations as failed to the optimizer performs better than reporting the mean of the folds that have been successfully evaluated for the MLP search space.

**Additional Experiment 2.** *Early stopping for **repeated** cross-validation.*

In Appendix C.2, we present results of ablating the effect of early stopping on 2-repeated 5- and 10-fold *inner* cross-validation. We conclude that the positive effect of early stopping cross-validation can also be found for repeated cross-validation. Moreover, 2-repeated 10-fold cross-validation improves, as expected, the generalization compared to just 10-fold cross-validation. Yet, surprisingly, while `Aggressive` previously never beat `No ES`, it does so in both cases, cf. Figure 14 and Figure 15, while even outperforming `Forgiving` in the 2-repeated 10-fold case.

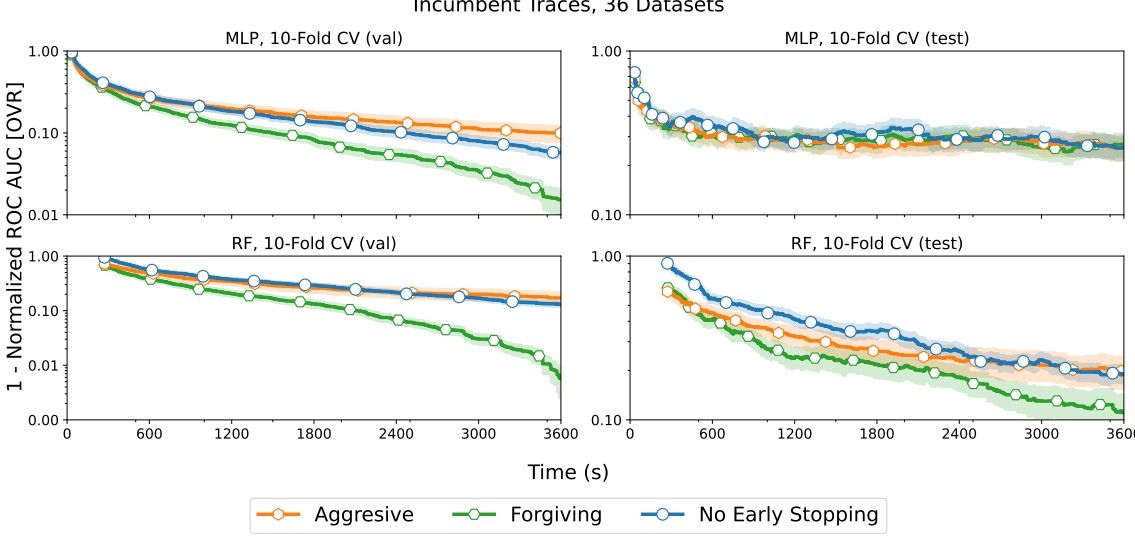

Figure 4: **Test and Validation Performance Comparison Example**: The validation (val) and test score incumbent trace for each method and the **10-fold** *inner* cross-validation scenario. The normalization of ROC AUC is explained in Appendix A.5. Note that the y-scales for (val) and (test) are different to improve readability.

## 5 Conclusion

We set out to make model selection with cross-validation more effective for AutoML. Therefore, we present an exploratory study that shows that *early stopping* of cross-validation can make model selection more effective across three cross-validation scenarios and 36 classification datasets. We find that `Forgiving` early stopping, a simple-to-understand and easy-to-implement method, (**1**) consistently allows model selection with random search to converge faster, in ∼94% of all datasets, on average by 214%; and (**2**) explore the search space more exhaustively by considering +167% configurations on average within a time budget of one hour; while also (**3**) obtaining better overall performance. Moreover, we show that early stopping can benefit model selection with Bayesian optimization and also *repeated* cross-validation.

We believe that our study reveals many promising avenues for future research to make model selection for AutoML more effective. We advocate for a renewed interest in the application of racing to adapt them to the requirements of state-of-the-art AutoML systems. Likewise, integrating early stopping of cross-validation natively into Bayesian optimization and assessing approaches to have the surrogate model learn from stopped and discard configurations seems promising.

Our work is limited by the scope of our study. We did not explore potential alternatives to cross-validation from multi-fidelity hyperparameter optimization. Furthermore, since we performed an abstract study, we did not implement and benchmark early stopping cross-validation in state-of-the-art AutoML systems.

**Broader Impact Statement**. Our work is mostly abstract and methodical. Thus, after careful reflection, we determined that this work does not present notable or new negative broader impacts that are not already present for existing state-of-the-art AutoML systems. We do, however, hope that our work contributes to making AutoML more computationally efficient and, thus, to making AI more sustainable.

**Acknowledgements**. Funded by the Deutsche Forschungsgemeinschaft (DFG, German Research Foundation) – Project-ID 499552394 – SFB 1597. This research was partially supported by TAILOR, a project funded by EU Horizon 2020 research and innovation programme under GA No 952215. Finally, we thank the reviewers for their constructive feedback and contribution to improving the paper.

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

## A Details on the Experimental Setup

### A.1 Datasets

See Table 3 for an overview of all selected datasets and their summary information.

Our selection of datasets was motivated mainly by ensuring that our exploratory study would not be biased due to our resource limitations. If we were to naively run our experiments on all datasets from the AutoML benchmark with the same resource constraints (primarily time budget being of importance), as is common practice (Gijsbers et al., 2022), then model selection would only have been able to evaluate a few configurations for a large subset of all dataset; biasing all our reported results. This bias would have been detrimental for our analysis w.r.t. convergence.

Therefore, we decide to select datasets for which model selection would evaluate a reasonably large number of configurations such that random search and Bayesian optimization would - to some extent - start converging. We set this number to 50 configurations, following the default value that is used within Auto-Sklearn to prune old models during model selection (Feurer et al., 2019).

Next, we measured how many configurations a random search can evaluate within a certain time constraint for the first fold of all datasets in the AutoML benchmark. Motivated by our computational constraints, we ran the MLP search space with 10-fold inner cross-validation without early stopping on a single CPU core with 4 hours of time and 5GB of memory.

Of the 71 original classification datasets, 31 were excluded, evaluating less than 50 configurations within the time budget. We excluded two datasets due to exceeding memory constraints. We excluded another two datasets (OpenML task IDs 2073 and 359974) for which the official AutoML Benchmark splits did not have all classes in one of their test splits. We ended up with a total of 36 datasets that were used for our study.

### A.2 Search Spaces

For our experiments, we use the `MLPClassifier` (MLP) and the `RandomForestClassifier` (RF) of `scikit-learn` (Pedregosa et al., 2011), with the search spaces adapted from Feurer et al. (2022). The spaces for each of these pipelines are provided in Tables 4 and 5. Four notable details of the pipelines we perform model selection over:

- We include feature processing as part of the pipeline, in such a way that all models sampled will work on all datasets we evaluate on. This includes the respective hyperparameters of the feature pre-processors.

- We fix the number of iterations for our pipelines such that sampled models are relatively consistent in terms of training time. The exception to this is that we also include the choice of using early stopping as part of the MLP pipeline.

- Entries prefixed with `cat:` indicate that this preprocessing component is only applied to categorical columns while `numerical:` indicates that it only applied to numerical columns.

- Our pipelines include conditional hyperparameters, such that a component such as `cat:OneHot` is chosen only if the corresponding indicator `cat:__choice__` is selected to be `OneHot`.

### A.3 Problems with Stratified Splitting and Our Solution

When using the outer splits as provided by the AutoML Benchmark Gijsbers et al. (2022), there are datasets where the training split of an official *outer* cross-validation fold has too few classes to allow for an *inner* $k$-fold stratified split, i.e., there are less than $k$ instances with a certain class label. To avoid this, we resample these underrepresented classes until we have $k$ such instances; allowing for stratified splitting. We sample without replacement unless this would not be enough to reach $k$ for a given class, in which case we sample with replacement. Our approach closely follows similar tricks employed by AutoGluon (Erickson et al., 2020) and Auto-Sklearn (Feurer et al., 2019, 2022).

Table 3: Supplementary information for all 36 selected datasets from the AutoML benchmark.

| Dataset Name | OpenML Task ID | #Instances | #Features | #Classes |
|---|---|---|---|---|
| Australian | 146818 | 690 | 15 | 2 |
| wilt | 146820 | 4839 | 6 | 2 |
| phoneme | 168350 | 5404 | 6 | 2 |
| credit-g | 168757 | 1000 | 21 | 2 |
| steel-plates-fault | 168784 | 1941 | 28 | 7 |
| fabert | 168910 | 8237 | 801 | 7 |
| jasmine | 168911 | 2984 | 145 | 2 |
| gina | 189922 | 3153 | 971 | 2 |
| ozone-level-8hr | 190137 | 2534 | 73 | 2 |
| vehicle | 190146 | 846 | 19 | 4 |
| madeline | 190392 | 3140 | 260 | 2 |
| philippine | 190410 | 5832 | 309 | 2 |
| ada | 190411 | 4147 | 49 | 2 |
| arcene | 190412 | 100 | 10001 | 2 |
| micro-mass | 359953 | 571 | 1301 | 20 |
| eucalyptus | 359954 | 736 | 20 | 5 |
| blood-transfusion-service-center | 359955 | 748 | 5 | 2 |
| qsar-biodeg | 359956 | 1055 | 42 | 2 |
| cnae-9 | 359957 | 1080 | 857 | 9 |
| pc4 | 359958 | 1458 | 38 | 2 |
| cmc | 359959 | 1473 | 10 | 3 |
| car | 359960 | 1728 | 7 | 4 |
| mfeat-factors | 359961 | 2000 | 217 | 10 |
| kc1 | 359962 | 2109 | 22 | 2 |
| segment | 359963 | 2310 | 20 | 7 |
| dna | 359964 | 3186 | 181 | 3 |
| kr-vs-kp | 359965 | 3196 | 37 | 2 |
| Bioresponse | 359967 | 3751 | 1777 | 2 |
| churn | 359968 | 5000 | 21 | 2 |
| first-order-theorem-proving | 359969 | 6118 | 52 | 6 |
| GesturePhaseSegmentationProcessed | 359970 | 9873 | 33 | 5 |
| PhishingWebsites | 359971 | 11055 | 31 | 2 |
| sylvine | 359972 | 5124 | 21 | 2 |
| christine | 359973 | 5418 | 1637 | 2 |
| wine-quality-white | 359974 | 4898 | 12 | 7 |
| Amazon_employee_access | 359979 | 32769 | 10 | 2 |

## A.4 Seeding

We ensure consistent seeding throughout our experiments to ensure both reproducibility and a consistent setting in which to compare methods. The root seed for each outer fold is given as $42 + \{0, 1, \dots, 9\}$, corresponding to the 10 outer folds provided by the AutoML Benchmark. This root seed is used to further seed various components in our evaluation pipeline.

1. **Inner cross-validation seeding**. This is given the same seed as the root seed.

Table 4: **Search Space MLP**: Hyperparameters which are fixed are designated as constant, while all other searched over hyperparameters are provided with their types and ranges.

| name | type | values | info |
|---|---|---|---|
| activation | category | relu,tanh | |
| alpha | float | $[1e\text{-}7, 0.1]$ | log |
| early_stopping | category | True,False | |
| hidden_layer_depth | integer | $[1, 3]$ | log |
| learning_rate | category | constant,invscaling,adaptive | |
| learning_rate_init | float | $[0.0001, 0.5]$ | log |
| momentum | float | $[0.8, 1.0]$ | log |
| num_nodes_per_layer | integer | $[16, 264]$ | log |
| numerical:SimpleImputer:strategy | category | mean,median | |
| cat:OrdinalEncoder:min_freq | float | $[0.01, 0.5]$ | log |
| cat:__choice__ | category | OneHot,passthrough | |
| cat:OneHot:max_categories | integer | $[2, 20]$ | log |
| max_iter | constant | 512 | |
| n_iter_no_change | constant | 32 | |
| validation_fraction | constant | 0.1 | |
| tol | constant | $1e\text{-}3$ | |
| solver | constant | adam | |
| batch_size | constant | auto | |
| shuffle | constant | True | |
| beta_1 | constant | 0.9 | |
| beta_2 | constant | 0.999 | |
| epsilon | constant | $1e\text{-}8$ | |
| cat:OrdinalEnc:unknown | constant | use_encoded_value | |
| cat:OrdinalEnc:unknown_value | constant | -1 | |
| cat:OrdinalEnc:missing | constant | -2 | |
| cat:OneHot:drop | constant | None | |
| cat:OneHot:handle_unknown | constant | infrequent_if_exists | |

2. **Optimizer seeding**: This is also given the same seed as the root seed, both for random search and when using SMAC for BO.

3. **Model seeding**: A model will be given a fixed seed from the optimizer, conditioned on the optimizers own seed. This fixed seed is used to create a `np.random.RandomState` which will generate $k$ different model seeds for the $k$ different folds on which the configuration will be evaluated on.

## A.5 Test Scores and Aggregation Across Datasets

**Test scores**. For each validated configuration, one model is trained on the training partition of each inner cross-validation fold and subsequently evaluated on the validation partition. For reporting test scores, we score the predictions of the $k$ fold models on the test partition of the *outer* fold, aggregated together with a bagged ensemble, where prediction probabilities are aggregated through *soft-voting*, that is, taking the mean of their probability outputs. This follows common practice in model selection for AutoML (Erickson et al., 2020; Feurer et al., 2022).

Table 5: **Search Space Random Forest Classifier**: Hyperparameters which are fixed are designated as constant, while all other searched over hyperparameters are provided with their types and ranges. We note that the parameter `max_features` is an ordinal hyperparameter spaced on a log scale with 10 values, not a continuous parameter.

| name | type | values | info |
|---|---|---|---|
| bootstrap | category | True,False | |
| class_weight | category | balanced,balanced_subsample,None | |
| criterion | category | gini,entropy | |
| max_features | ordinal | logspace(0.1, 1, n=10) / 10 | |
| min_impurity_decrease | float | $[1e\text{-}09, 0.1]$ | log |
| min_samples_leaf | integer | $[1, 20]$ | log |
| min_samples_split | integer | $[2, 20]$ | log |
| numerical:SimpleImputer:strategy | category | mean,median | |
| cat:__choice__ | category | OneHot,passthrough | |
| cat:OneHot:max_categories | integer | $[2, 20]$ | log |
| n_estimators | constant | 512 | |
| max_depth | constant | None | |
| min_weight_fraction_leaf | constant | 0.0 | |
| max_leaf_nodes | constant | None | |
| cat:OrdinalEnc:unknown | constant | use_encoded_value | |
| cat:OrdinalEnc:unknown_value | constant | -1 | |
| cat:OrdinalEnc:missing | constant | -2 | |
| cat:OneHot:drop | constant | None | |
| cat:OneHot:unknown | constant | infrequent_if_exists | |

**Aggregation**. When aggregating scores, we must apply some normalization for meaningful aggregate statistics. To do so, we construct the *incumbent trace* for each method on each outer fold of each dataset. This is the cumulative maximum of the best configuration as given by the validation score. We then take the mean across the 10 outer folds to obtain a mean incumbent trace for every method on every dataset. To normalize performance ranges across datasets, we first ensure that we only take a mean at the first time point at which there is an evaluated point for every dataset. We then apply a min-max normalization to each dataset, taking the minimum and maximum as seen by each mean incumbent trace for that dataset, inverting the values to obtain the regret for each method from the best value seen. Lastly, we compute the mean and standard error from the mean as our aggregate statistics over datasets which we use throughout the plots we present. When applying aggregations to test scores, we take the test score of the configurations plotted along the incumbent trace, however treating test scores as their own population for the purposes of min-max scoring.

## B  Supplementary Results

### B.1  [Hypothesis 1] Additional Dataset-specific Speedup Plots

In the following section, we show speedups for the MLP search space with 3-fold (Figure 5) and 5-fold (Figure 6) cross-validation, to better see the speedup behavior of both `Aggressive` and `Forgiving` across scenarios. We also include speedups for the RF search space with 10-fold cross-validation (Figure 7, where surprisingly, we see that the cases where `Aggressive` fails to reach the best performance of no early stopping, are *all* datasets where `Aggressive` failed for the MLP search space with 10-fold cross-validation.

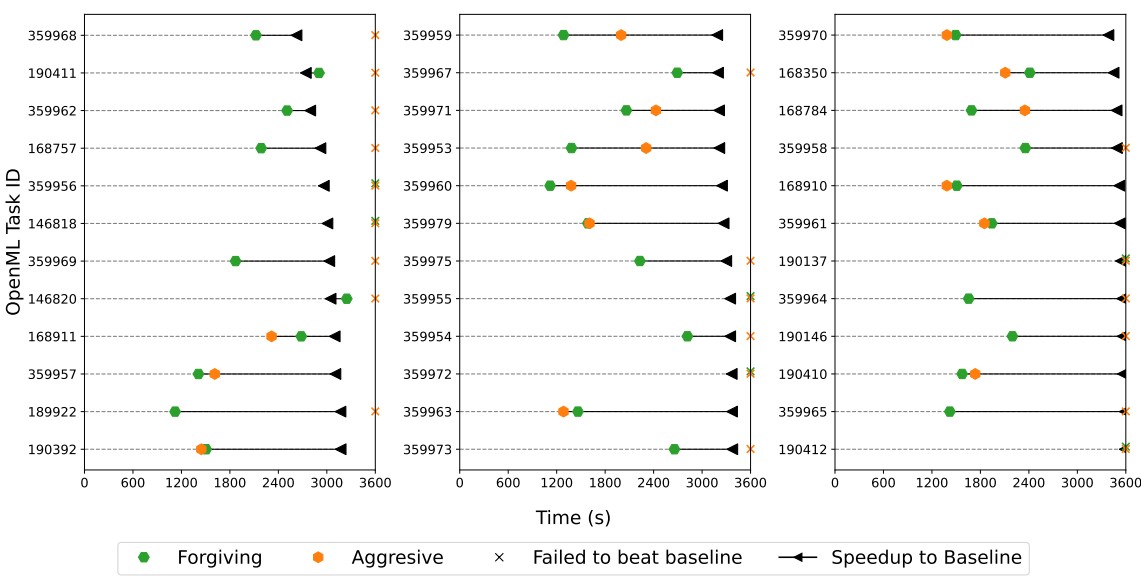

Figure 5: **Speedups for MLP with 3-fold**: Please see Figure 1 for a description of the figure. While the time to best performance of No ES is notably more delayed than shown in Figure 1, we also show in Figure 8 that no early stopping has a significantly increased sampling density due to less folds to evaluated. With fewer folds from which to save evaluation time, both Forgiving and Aggressive show less impressive speedups with failure cases for Forgiving, non-existent with in 10 fold cross-validation.

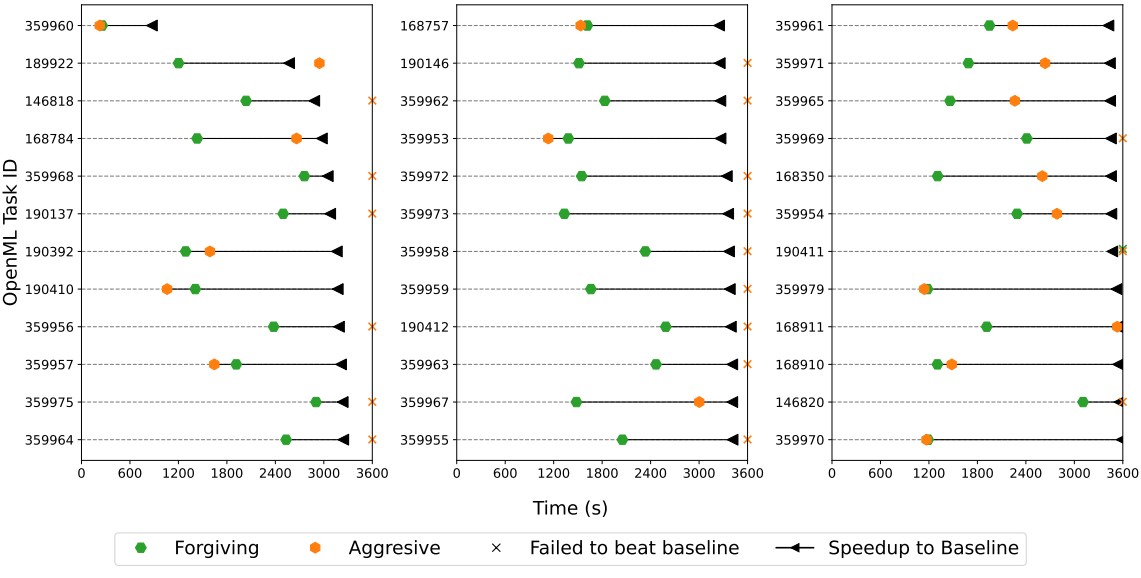

Figure 6: **Speedups for MLP with 5-fold**: Please see Figure 1 for a description of the figure. We find that the characteristics of the speedups in this setting do not deviate much from that of 3- and 10- fold cross validation.

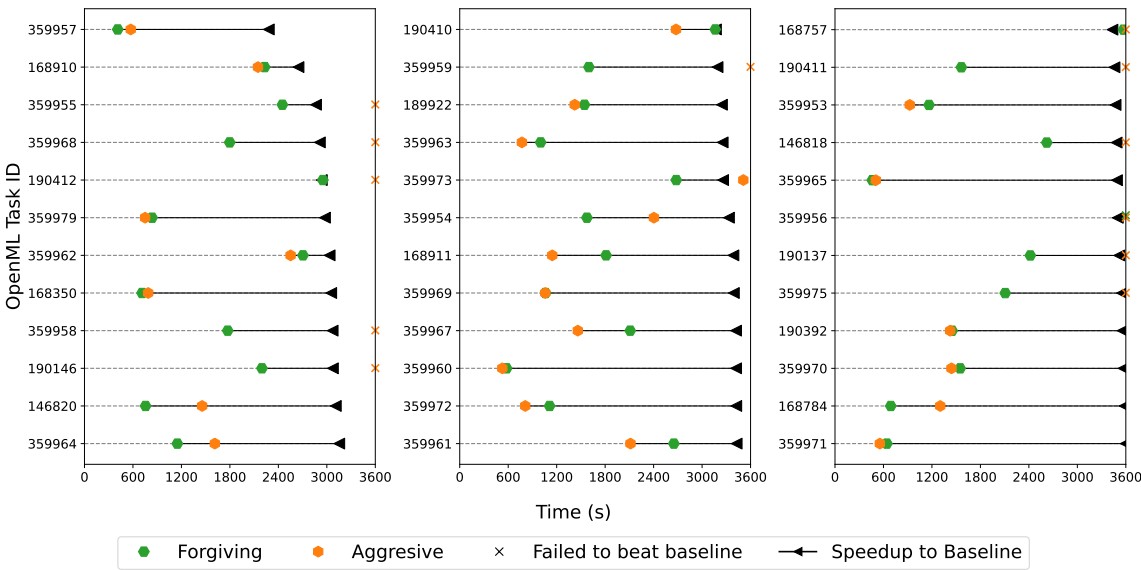

Figure 7: **Speedups for RF with 10-fold**: Please see Figure 1 for a description of the figure. Of the 12 datasets where `Aggressive` failed to outperform no early stopping in this scenario, *all* of these datasets are also instances where `Aggressive` failed for the MLP pipeline with 10 folds, indicating that datasets and quality of splits may play a larger role than that of pipeline space being searched over.

## B.2 [Hypothesis 2] Additional Footprint Plots

See Figure 8 for an extended example of footprint plots for the MLP search space. This figure shows the density sampling of no early stopping and both methods as the number of folds considered increases.

## B.3 [Hypothesis 3] Additional Test and Validation Performance Comparisons

See Figure 9 and Figure 10 additional test and validation performance comparisons for 3- and 5-fold evaluation of MLP and RF.

## C Additional Experiments

### C.1 [Additional Experiment 1] *Early stopping cross-validation of model configurations; replacing random search with Bayesian optimization (BO).*

In this section, we investigate the impact of replacing random search with Bayesian optimization (BO) as the model selection method, to observe the performance of early stopping cross-validation of model configurations. We mirror the experimental design described in Section 3.2 and show in Figures 11, 12, 13, the incumbent trace and speedup plots used in Section 4.

As described in Section 3.2, we investigate two variants of BO: reporting a discarded configuration as failed using the worst possible score (*failed*), or reporting the mean of the folds that have been successfully evaluated (*mean*).

To extend on the discussion from Section 3.2, we would like to note that reporting configurations as failed can lead to discontinuities in the landscape of the probabilistic model, which can lead to bad probabilistic estimates in many cases. On the other hand, reporting the mean of the folds that have been successfully evaluated could misguide the BO surrogate model into believing that some regions of the configuration space are poorly performing.

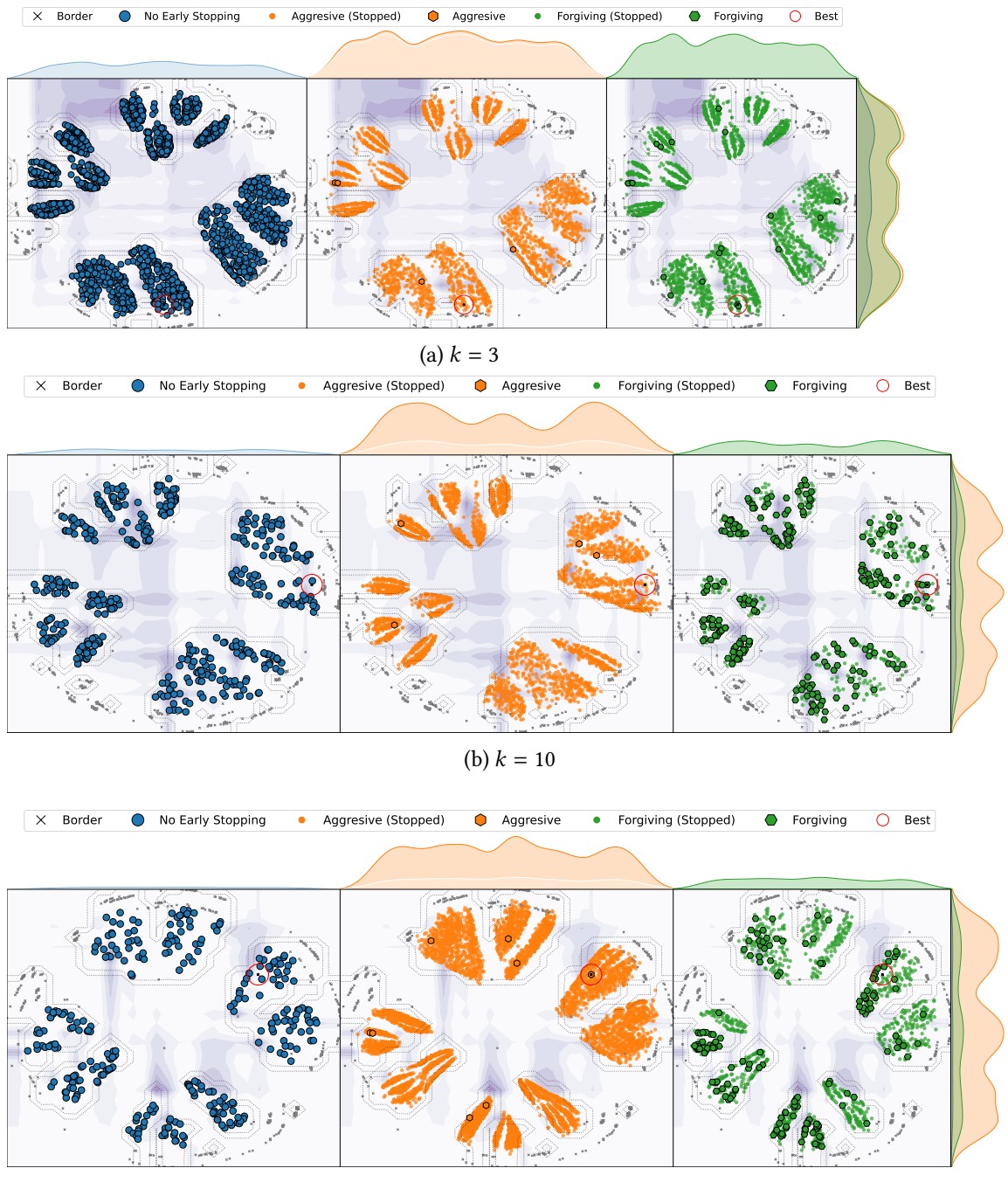

(a) $k = 3$

(b) $k = 10$

(c) 2-repeated 10-fold

Figure 8: **Footprint Plot Example: This example is for OpenML Task ID** $355964$ **on outer fold** $0$ **with** $k = 3$ **(top),** $k = 10$ **(middle) and 2-repeated 10-fold (bottom)** Please refer to Figure 2 for more description of the figure. Looking at no early stopping (left), we see that as the number of fold increases, the sampling density rapdily decreases for no early stopping, as it takes less time to evaluate a given configuration. However, for both `Aggressive` and `Forgiving`, the density drop is not as dramatic, with the notable observation that `Forgiving` early stops significantly more configurations for 3 fold, similar to that of `Aggressive`, while allowing more configurations to continue as the fold count increases. Lastly, despite the very few full evaluations of `Aggressive` in 2-repeated 10-fold, it does manage to locate the best configuration.

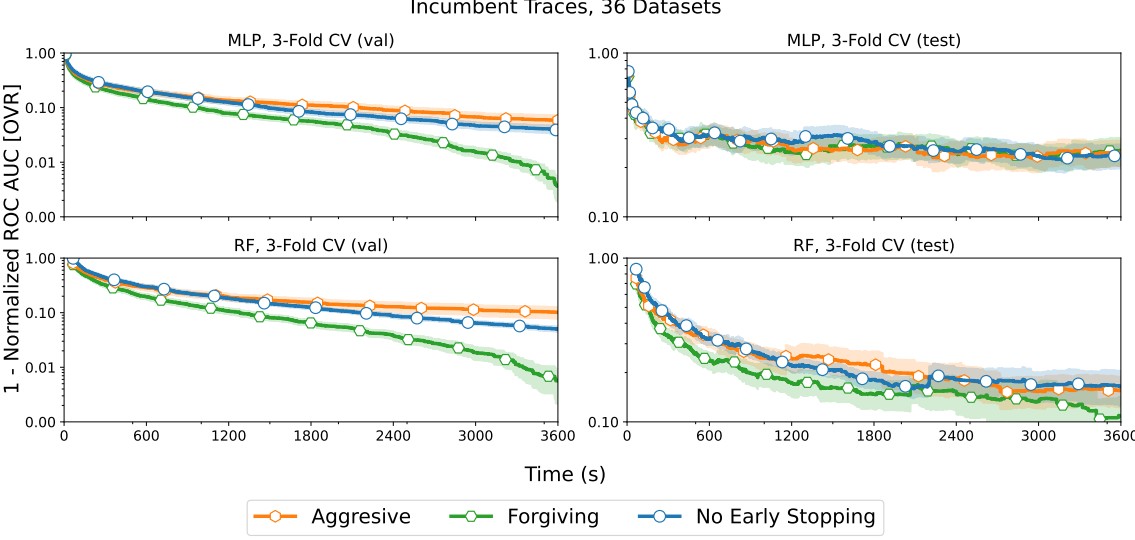

Figure 9: **Test and Validation Performance Comparison for 3 fold Cross-validation**: The validation (val) and test score incumbent trace for each method. The normalization of ROC AUC is explained in Appendix A.5. Note that the y-scales for (val) and (test) are different to improve readability.

In Figure 11, we observe that `Forgiving` beats `No ES` (that is, no early stopping), but `Aggressive` does not. Moreover, `Aggressive` performs much worse than the other two approaches. Another notable conclusion is that reporting the discarded configuration as failed can be beneficial to the final aggregated performance, especially for the MLP search space. For the RF search space, reporting the mean of evaluated folds, is marginally preferred but inconclusive. These conclusions are only observable in the later time span of optimization, within which a BO method is more exploitative after having observed points throughout the landscape.

### C.2 [Additional Experiment 2] Early stopping for repeated cross-validation

In this additional experiment, we investigate the impact of replacing the *internal* cross-validation with repeated cross-validation on the performance of early stopping. We mirror the experimental design described in Section 3.2 and the method to generate plots used in Section 4.

In Figure 15, 14, we present the results for using 2-repeated *inner* cross-validation. Here, `Aggressive` consistently beats `No ES` in validation performance with a slight test score advantage for 2-repeated 10-fold. Moreover, `Aggressive` beats `No ES` in both scenarios, even matching `Forgiving` for 2-repeated 10-fold in the MLP search space. `Forgiving` dominates the RF search spaces and has a particularly large gap in validation performance compared to `No ES`, which does not fully generalize to test performance.

We also show additional speedup plots in Figure 16.

## D  Supplementary Experiments

### D.1  Empirical Evidence of Ranking Correlation

As stated at the end of Section 3.1, our intuition on why early stopping methods are able to perform better than no early stopping relies on the assumption that the average over a subset of all fold evaluations is representative of the configuration's average performance over all fold evaluations. This assumption implies that the ranking of partially evaluated configurations up to the fold $n$ should remain relatively consistent with their rankings once we evaluated all $k$ folds.

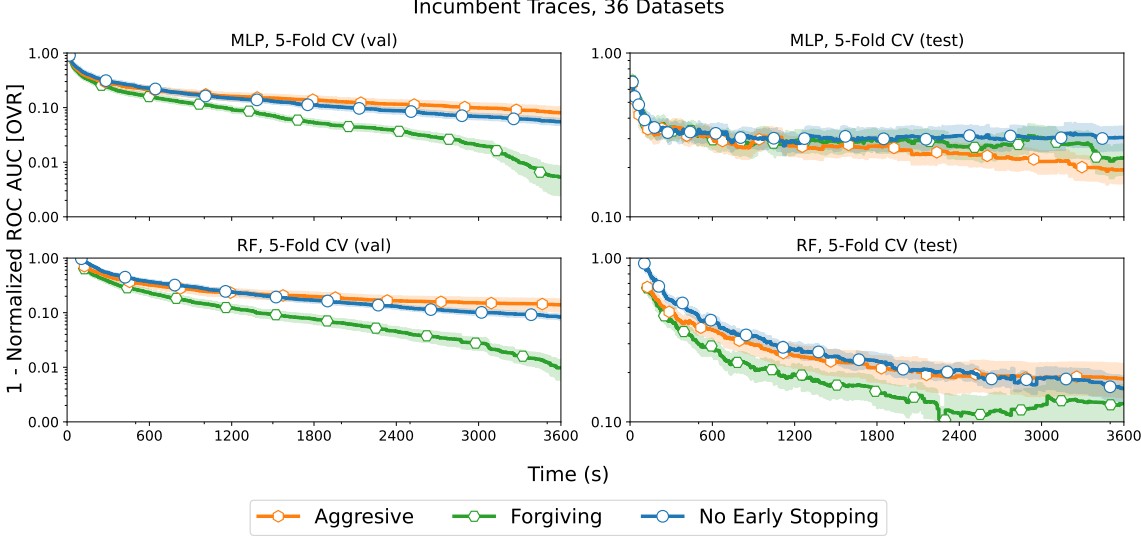

Figure 10: **Test and Validation Performance Comparison for 5 fold Cross-validation**: The validation (val) and test score incumbent trace for each method. The normalization of ROC AUC is explained in Appendix A.5. Note that the y-scales for (val) and (test) are different to improve readability.

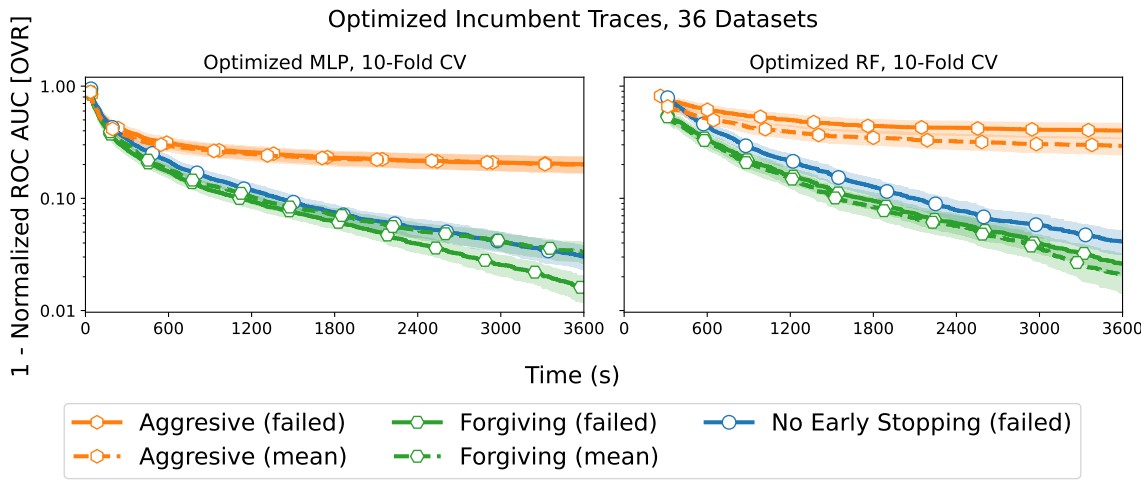

Figure 11: **Validation Performance Over Time for Bayesian Optimization (BO)**: The validation score incumbent trace for each early stopping method, the 10-fold cross-validation scenario, and when using BO as model selection strategy. Additionally, the different strategies for reporting the performance of discarded configurations to the optimizer are depicted using different line styles. The normalization of ROC AUC is explained in Section 3.2.

To further investigate this, we reuse the data collected for the random search with 10-fold cross-validation scenario from our experiments in Section 4. We use the performance of the configurations from the MLP search space evaluated during model selection without early stopping to have the performance of each configuration for each of the $k = 10$ inner folds. We then calculate the rankings of the configurations given their average performance per subset of folds up to and including fold $n$ for $n \in \{1, ..., k\}$. Finally, we calculate the Spearman's rank correlation between the rankings for $n$ folds with the rankings for $k$ folds.

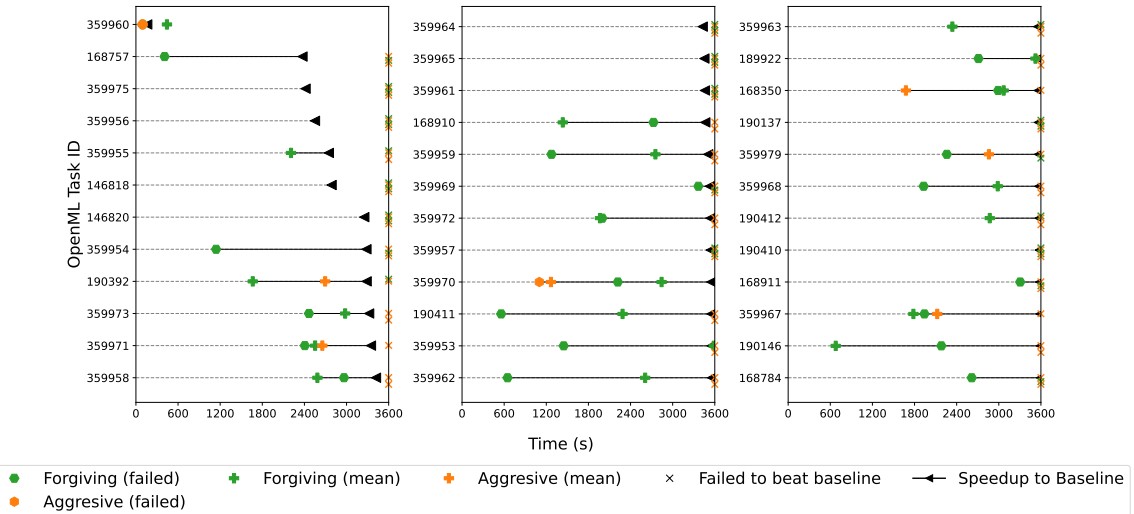

Figure 12: **Speedup for BO optimized MLP, 10 fold**: Please see Figure 1 for a description of the figure. Comparatively to 1, we see that there are a lot more datasets where `Aggressive` variants fail to match the performance of `No ES`, while `Forgiving` remains fairly consistent in achieving a speedup. We can see that the results shown here corroborate with Figure 11, in that for the MLP space, reporting early stopped configurations as failed achieves greater speedups than reporting the mean of evaluated folds.

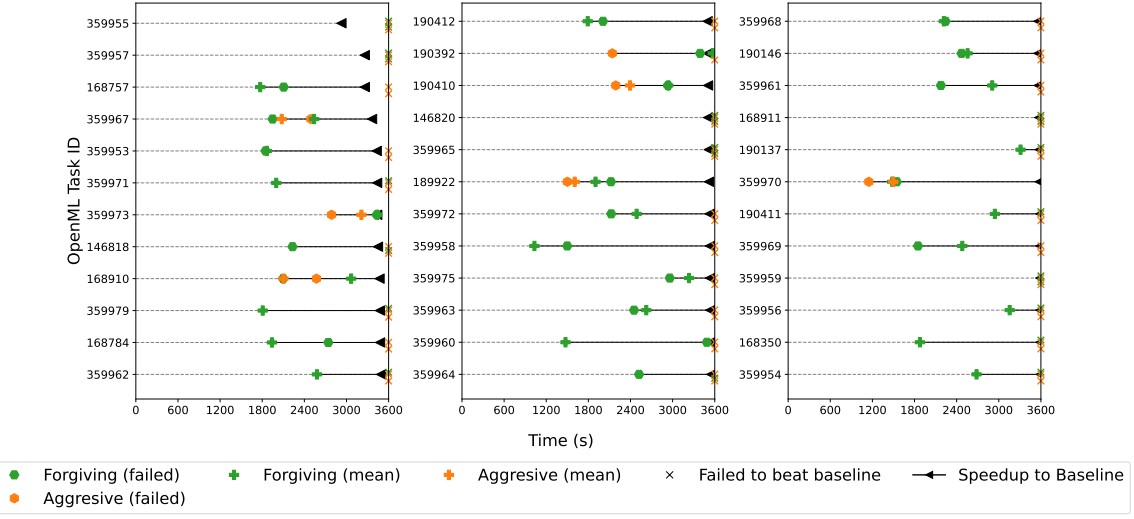

Figure 13: **Speedup for BO optimized RF, 10 fold**: Please see Figure 1 for a description of the figure. Comparatively to 1, we still see that `Aggressive` variants fail to match the performance of `No ES`, while `Forgiving` is consistent in achieving some speedup. Based on the incumbent traces displayed in Figure 11, we see that there is no clear strategy on what to observe for early stopped configurations, however `Forgiving` is still more consistent in achieving some form of speedup.

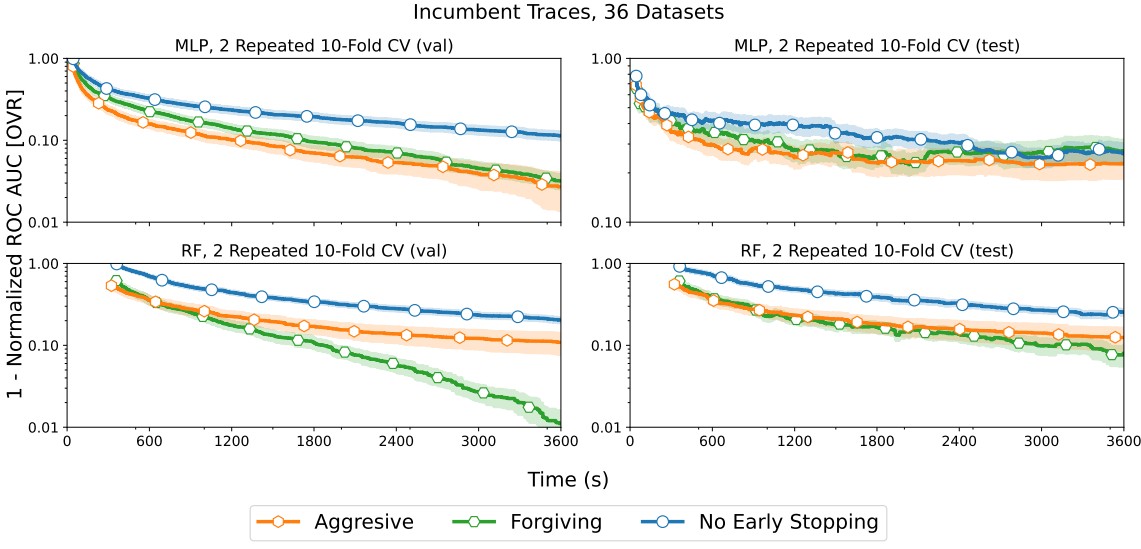

Figure 14: **Test and Validation Performance Comparison for 2-Repeated 5-Fold Cross-validation**: The validation (val) and test score incumbent trace for each method. The normalization of ROC AUC is explained in Appendix A.5. Note that the y-scales for (val) and (test) are different to improve readability.

Figure 15: **Test and Validation Performance Comparison for 2-Repeated 10-Fold Cross-validation**: The validation (val) and test score incumbent trace for each method. The normalization of ROC AUC is explained in Appendix A.5. Note that the y-scales for (val) and (test) are different to improve readability.

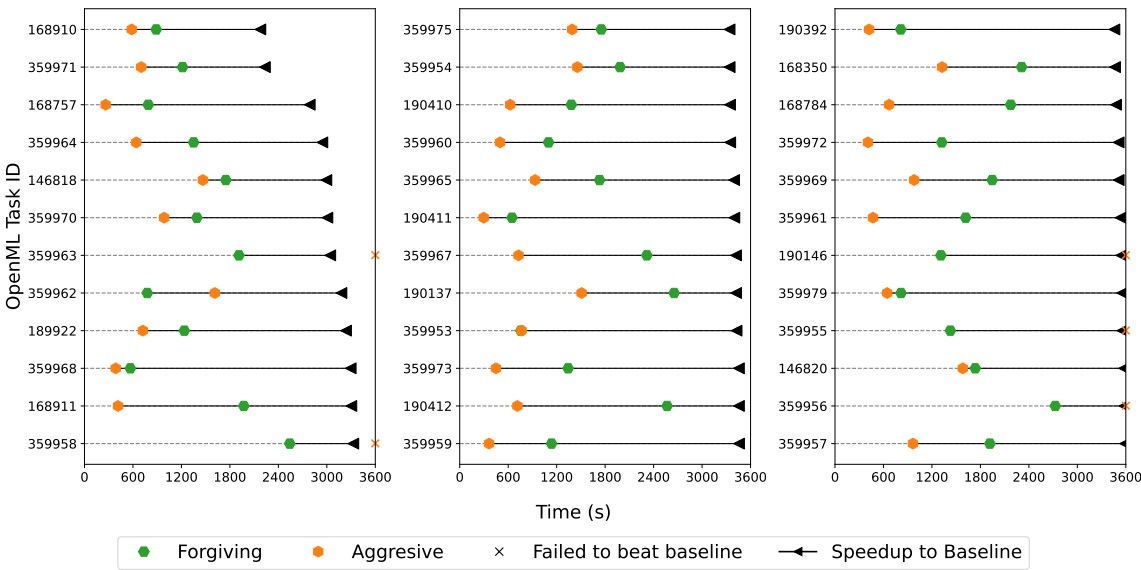

Figure 16: **Speedups for MLP with 2-repeated 10-fold**: Please see Figure 1 for a description of the figure. When considering more total folds, and hence more possible time to save, the speedups obtained by Aggressive are consistently greater than Forgiving with notably less failure cases than the results shown for Figure 1, which featured the $k = 10$ case.

We show the results in Figure 17. We observe that a high correlation exists for all compared datasets. The dataset with the lowest observed correlation has a minimum correlation of ∼0.5 between the first and last fold. In other words, even in the worst case, the rankings are reasonably correlated. All other datasets display a stronger correlation than this for all $k$ folds.

## D.2 A Happy Medium between Aggressive and Forgiving: Robust-3/5

While investigating both the Aggressive and Forgiving exploratory methods, a natural assumption would be that some middle ground between the two could perhaps outperform both.

In Figure 18, we show the results for Robust 3 and Robust 5, two such variants of a method between Aggressive and Forgiving. Although Robust 5 is quite beneficial in the MLP search space, the same cannot be said for the RF search space.

To balance being less aggressive than Aggressive and less forgiving than Forgiving, Robust 3 and Robust 5 rely on maintaining a population of the top $M$ configurations, e.g. $M = 3$ for Robust 3 and $M = 5$ for Robust 5. Given the population, we define a criterion by which a configuration is allowed to continue if it improves the quality of the population.

In particular, we measure the quality of the population by the statistic $\mu_M - \sigma_M$ based on the mean $\mu_M$ and standard deviation $\sigma_M$ of the population's cross-validation scores. Then, when cross-validating a configuration $c$, after each fold, we check the potential value of the statistic if we were to replace the fold score of some member in the population with the average of the scores across all so far evaluated folds for $c$. We allow $c$ to continue cross-validation, if the potential value of the statistic is better - i.e., higher. A potential configuration can improve the statistics by either shifting $\mu_M$ to a higher value or shrinking the deviation.

Every fully evaluated configuration will replace a member of the population. This always holds because only configurations that would improve the population are fully evaluated and not stopped early.

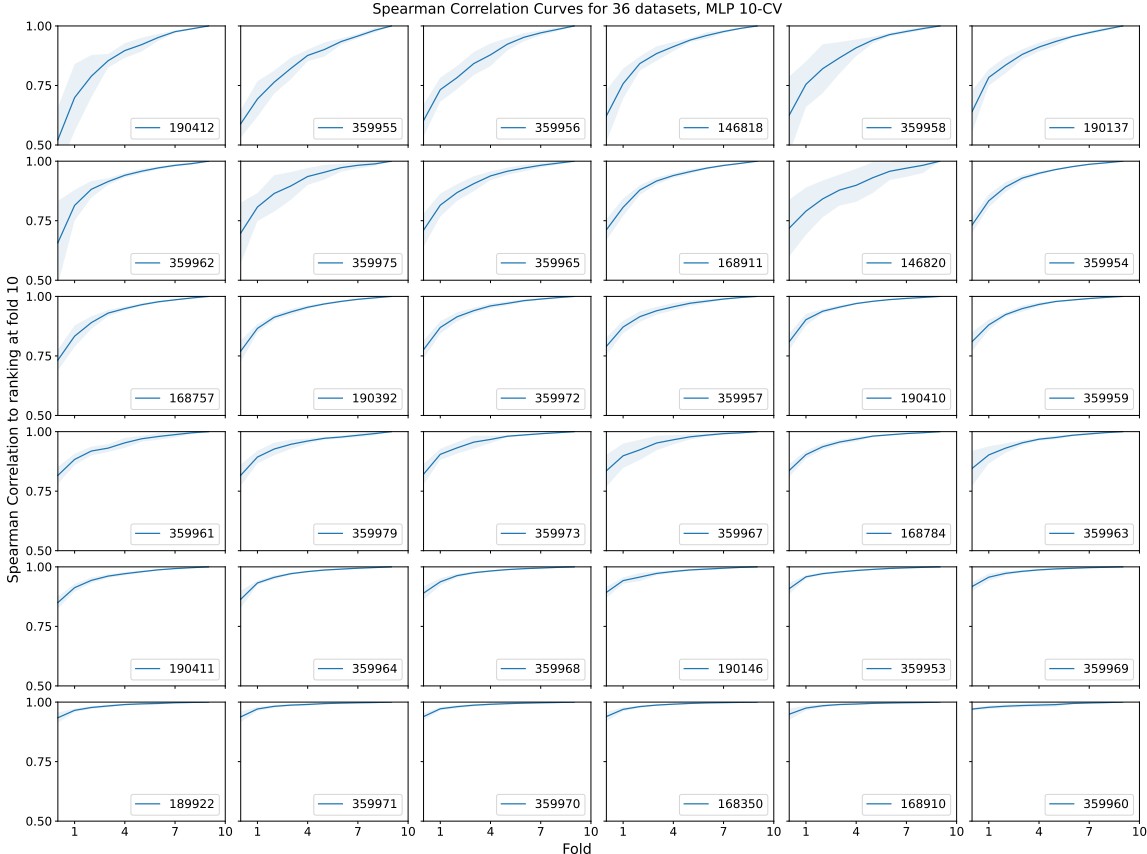

Figure 17: **Spearman's Rank Correlation Curves:** The Spearman's rank correlation between the ranking of MLP configurations evaluated during random search after a given fold (the x-axis) to the ranking of those same configurations once fully evaluated, i.e. at fold 10, for all 36 datasets. The configurations are ranked based on their average ROC AUC, up to and including fold *n*. Consequently, the correlation at fold 10 is always 1, as it is the correlation between two identical rankings. The blue line represents the mean, and the shaded area represents the standard deviation of the correlation curves. The mean and standard deviation are taken across all 10 outer folds for a given dataset. The subplots are ordered according to the correlation observed at fold 1, with the lowest being in the top left and the highest correlation in the bottom right.

## E Misc

### E.1 Discussion on the Adaption of Existing Methods for Early Stopping to Cross-Validation in AutoML Systems for Tabular Data

During the initial research for the exploratory study presented in the main paper, we found it very surprising that early stopping has not been applied more often *to stopping cross-validation* in AutoML systems for tabular data. To date, Auto-Weka (Thornton et al., 2013) has been the only popular AutoML system for tabular data that employs early stopping of cross-validation. Given the long-standing literature on early stopping techniques from the fields of racing and multi-fidelity optimization and the obvious improvement in efficiency when early stopping cross-validation for tabular data, we expected it to be much more prominent. Furthermore, using early stopping techniques from racing and multi-fidelity optimization is not unheard of in AutoML systems for tabular data. Auto-Sklearn 2 (Feurer et al., 2019, 2022) included the multi-fidelity

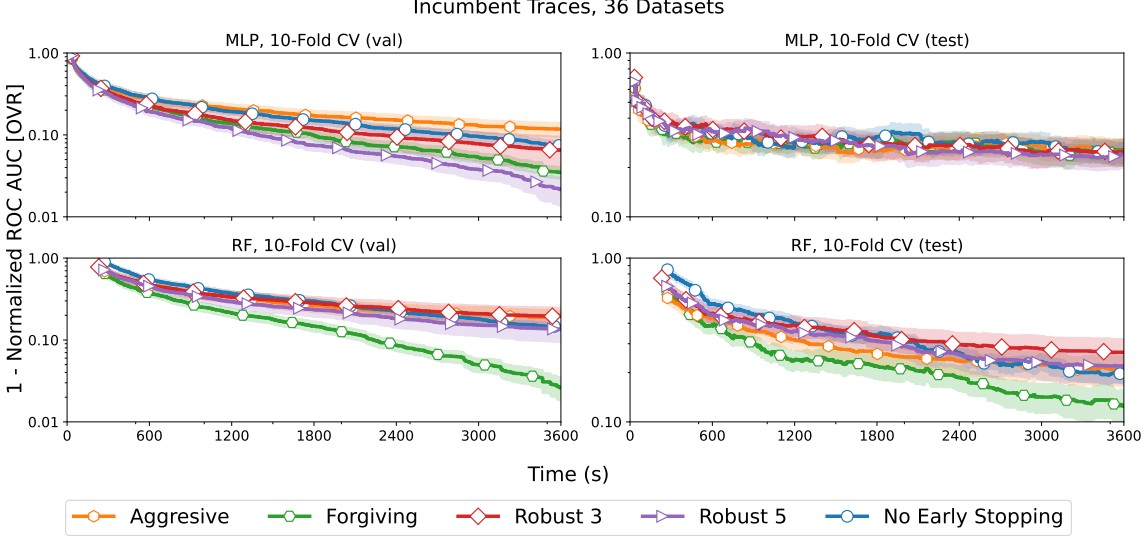

Figure 18: **Test and Validation Performance for 10 fold Cross-validation with Robust 3/5**: The validation (val) and test score incumbent trace for each method. The normalization of ROC AUC is explained in Appendix A.5. Note that the y-scales for (val) and (test) are different to improve readability.

technique Successive Halving (Jamieson and Talwalkar, 2016) to early stop the sequential training of models; and Auto-Weka recently won a test of time award. Likewise, the general field of AutoML is seemingly fully aware of racing and multi-fidelity optimization, as exemplified by the AutoML book (Hutter et al., 2019), which prominently mentions both. Yet, as of the most recent AutoML Benchmark report (Gijsbers et al., 2022), none of the state-of-the-art AutoML systems for tabular data employ early stopping of cross-validation.

To the best of our knowledge, there is no documented cause or theoretical limitation that hindered the adaption of prior work on early stopping for cross-validation in state-of-the-art AutoML systems. Thus, we hypothesize that *practical* concerns are the main reasons for hindering their adaption. To shed light on this for future researcher and developer of AutoML systems, we present a short discussion of practical challenges when considering to adapt algorithms from multi-fidelity optimization or racing to early stopping of cross-validation in AutoML systems for tabular data:

**Multi-Fidelity Optimization**. HyperBand (Li et al., 2018), one of the most prominent multi-fidelity algorithms, could be used for early stopping cross-validation by treating folds as the fidelity. This and related algorithms dynamically allocate resources to configurations deemed to be more promising in terms of end performance. Dynamic resource allocation requires that configuration can be *paused*, to be resumed later if it is better than competitors in its bracket. Thus, dynamic resource allocation necessitates that we must be able to *pause* a configuration until the full bracket has been evaluated. How to implement dynamic resource allocation is a very practical problem when extending an AutoML system. If care is not taken, many silent issues may arise, e.g. memory consumption, repeated expensive reads of a dataset or models from disk, sub-optimal fold-wise parallelization, or inefficient scheduling w.r.t. a time budget. While none of these are a theoretically limitation to modern variants of multi-fidelity algorithms (e.g., see ASHA (Schmucker et al., 2021)), this is a big ask of any existing state-of-the-art AutoML systems. Albeit, new AutoML systems could harness the power of being fully compatible with multi-fidelity-based methods; which might not be trivial given that it has not been done so far.

**Racing.** Racing can be used for early stopping (cross-)validation (Maron and Moore, 1997). Similar to multi-fidelity optimization, racing faces the same issues related to pausing configurations because racing also dynamically allocates resources, i.e., what to race against each other. In addition, racing faces practical concerns from applied statistical testing. Of particular relevance is that packages such as iRace, (López-Ibáñez et al., 2016) which employs statistical testing to determine the more promising configuration, are often used in a setting with a possibly non-finite instance set from which samples are drawn. It is not entirely clear how statistical testing performs in a scenario such as 3 fold CV, where statistics such as the Friedmann test will fail to refute the null hypothesis. Racing also typically assumes that samples are *i.i.d.*, but this is not the case in cross-validation. To reiterate, all of these issues can likely be fixed or evaded; however, when building a new or extending an AutoML system, our community first needs to determine in what way this could be done.

**Conclusion.** To summaries, we hypothesize that multi-fidelity and racing algorithms face *practical* concerns that have hindered and will further hinder their adaption for existing AutoML systems. In contrast, we believe that for AutoML systems to re-engage with related literature for early-stopping, methods explored in research must first meet existing AutoML systems where they are at, to show that there are easy benefits to be gained from even the most simple and adaptable methods. In other words, we first need to investigate easy-to-implement methods, e.g., they do not require *pausing* of a configurations. In conclusion, the focus of this exploratory study on simple-to-understand and easy-to-implement methods for early stopping cross-validation is partially[3] motivated by the currently missing and likely further hindered adaption of existing methods from multi-fidelity optimization and racing.

## E.2 Code to Reproduce the Experiments

We release our code under a BSD-3-Clause license, provide documentation, install scripts, our raw results, and a script to reproduce a minimal example of our experiments. Our code is available on GitHub: `https://github.com/automl/DontWasteYourTime-early-stopping`.

## E.3 Total Amount of Compute

We estimate our total compute time for our experiments to be ~6.15 CPU years.

We estimated the number as follows. We had to run model selection for 1 hour on 4 CPU cores for 10 folds for 36 datasets. We had to run model selection with these constraints for 4 cross-validation scenarios (3-fold, 5-fold, 10-fold, 2-repeated 10-fold) and 3 methods (`Aggressive`, `Forgiving`, No Early Stopping) and 2 search spaces (MLP, RF). In addition, we had to run the model search for BO with the same constraints for 5 methods, 1 cross-validation scenario (10-fold), and 2 search spaces (MLP, RF). We add 10% as an estimate for overhead. Thus, we obtain $(1 * 4 * 10 * 36) * ((4 * 3 * 2) + (5 * 1 * 2)) * 1.1 = 53856$ CPU hours, or ~6.15 CPU years of compute.

## E.4 Used Assets

We used the following existing assets to enable our research:

- scikit-learn (Pedregosa et al., 2011), BSD-3-Clause License, Version 1.4.1.post1

- AMLTK (Bergman et al., 2023), BSD-3-Clause License, Version 1.11.0

- Openml-python (Feurer et al., 2021), BSD 3-Clause License, Version 0.14.2

- scipy (Virtanen et al., 2020), BSD-3-Clause License, Version 1.12.0

---

[3]The focus on on simple-to-understand and easy-to-implement methods is further motivated by reducing confounding factors, see the main paper.

- SMAC (Lindauer et al., 2022), BSD-3-Clause License , Version 2.0.2

- seaborn (Waskom, 2021), BSD-3-Clause License , Version 0.13.2

- matplotlib (Hunter, 2007), Custom, Version 3.8.3

- pandas (McKinney, 2010), BSD-3-Clause License, Version 2.2.1

