# OpenReview forum: "Don't Waste Your Time: Early Stopping Cross-Validation"
_automl.cc/AutoML/2024/Conference — AutoML 2024_

### Official Review · Reviewer_4SE8 · 2024-03-24

**Potential Impact On The Field Of Automl Rating:** 3
**Technical Quality And Correctness Rating:** 3
**Clarity:** The paper is well written and has a c…
**Clarity Rating:** 4

**Summary Of Contributions:**

The paper targets speeding up the process of model selection via $k$-fold cross-validation by stopping the calculations early if the first folds procedure sufficiently bad results.
The authors propose two simple methods, one "aggressive" and the other "forgiving" with respect to how early a cross-validation calculation should be terminated.
The paper includes extensive experiments to evaluate the effectiveness of the proposed methods, looking at the resulting speedup in random search and Bayesian optimization.
A thorough comparison shows that both methods can offer significant speedup in model selection via saved time and more effective exploration of the model space, but the forgiving, more conservative variant yields safer, more robust results.

**Actions Required To Increase Overall Recommendation:**

Perhaps in the process of conducting the study, the authors have already investigated what I think could serve as a reasonable compromise between the two extremes proposed in the paper.
Any discussion on this would be appreciated!

**Overall Review:**

The paper tackles an important problem that is relevant among the AutoML community.
As mentioned in the text, while the practice of early stopping has been studied before, it is still not widespread, and this work makes an effort to attract more attention and make early stopping more applicable.
The experiment section in particular is thorough and convincing in showing the effectiveness of the methods.
As indicated by the paper, the application of the proposed methods can potentially lead to more sufficient model selection in machine learning.

As mentioned above, I do think the authors could have taken an extra step in deriving a reasonable early stopping rule that balances between aggressiveness and conservativeness.
Given the computational and time cost involved in tuning models' hyperaparameters mentioned in the paper, a more efficient yet still sufficiently safe strategy could prove quite useful.

**Potential Impact On The Field Of Automl:**

The paper's potential impact on the field is significant.
I would likely cite the paper.

**Review Confidence:**

3

**Review Rating:**

8

**Review Summary:**

The paper studies an important problem and proposes strategies that are shown to be effective.
While I think that more effort could have been made in designing a more effective early stopping strategy, this work serves as an important step towards studying the practice of early stopping.
I therefore lean towards acceptance.

**Technical Quality And Correctness:**

While the proposed methods are simple in their formulations, the experiments are extensive and offer insightful conclusions to the study.
While the authors fully acknowledge the simplicity of the methods and state that even the presented variants yield significant value, I do feel that it was a missed opportunity that a "happy medium" between the aggressive and forgiving strategies wasn't looked into.
It seemed to me by deriving some simple decision rule with probabilistic confidence bounds, one can come up with a strategy that is more aggressive than the presented forgiving one but is still conservative enough to not miss out on good solutions.

---

### Official Review · Reviewer_Pk7G · 2024-03-27

**Potential Impact On The Field Of Automl Rating:** 4
**Technical Quality And Correctness Rating:** 4
**Clarity Rating:** 3

**Summary Of Contributions:**

&nbsp;

The authors investigate the impact of early stopping cross-validation on two benchmark problems involving hyperparameter optimisation of MLP and Random Forest classifiers respectively. The motivation behind the study is that cross-validation is a time-intensive procedure and often there exists early signal that an evaluated model configuration (hyperparameter setting) is not performant relative to the incumbent best configuration observed so far. This observation is somewhat related to the optimization of multifidelity objective functions where the argument is that an objective that is of low fidelity, but less costly to evaluate can be a good proxy for the original, high fidelity objective function. The authors examine two forms of early stopping, aggressive and forgiving which trade-off the point at which the decision to stop early is made. Additionally, the authors provide additional experiments examining the performance of early stopping for repeated cross-validation and for Bayesian optimization as the algorithm for configuration selection.

&nbsp;

**Actions Required To Increase Overall Recommendation:**

&nbsp;

Principally the improvements in clarity mentioned above!

&nbsp;

**Clarity:**

&nbsp;

1. In the abstract, it may be worth rephreasing the BO and repeated cross-validation ablations as additional experiments or similar since they are not explicitly evaluating the effect of removing or adding a component but rather replacing a component.

2. When stating the statistic of 167% in the abstract, it would be worth emphasisng that this is subject to a time budget.

3. In the introduction, it would be worth explicitly mentioning that a configuration corresponds to a hyperparameter setting.

4. In the introduction, where does the factor of 12x arise? Is this not greater than linear rather than "almost linear" in the number of folds? Is it possible to state the complexity in big-oh notation i.e. O(N)?

5. In the introduction the connection to Occam's Razor is somewhat unclear.

6. In the introduction, "Racing" is mentioned before it is defined.

7. Line 103, "is not straightforward"?. Line 104, "is not applicable" present tense.

8. Line 113, 2015 is not that recent!

9. In Section 3.1 should F_{cj} have the configuration j included on the RHS?

10. Line 135 missing caligraphic N for the set of natural numbers?

11. It would be great if the inner and outer cross-validation loops could be explained in more detail in the main paper. To this end the information from Section A5 of the appendix could be included in the main paper.

12. Page 6, the acronym MDS could be defined as multidimensional scaling.

13. In the caption of table 1, the reported errors can be described.

14. In Figure 2, what is the estimated boundary of viable configurations?

15. Line 549 typo.

16. It may be worth differentiating the problem setting here relative to cross-validation considered in the BO community as a means of validating the surrogate model in BO (as opposed to the model over which HPO is being performed in AutoML). An example of this from the BO literature is the Efficient Global Optimization (EGO) algorithm in [1].

&nbsp;

**__References__**

&nbsp;

[1] Jones, D.R., Schonlau, M. and Welch, W.J., 1998. [Efficient global optimization of expensive black-box functions. Journal of Global optimization](https://link.springer.com/article/10.1023/A:1008306431147), 13, pp.455-492.

&nbsp;

**Overall Review:**

&nbsp;

The contribution of the submission is to highlight the potential for early stopping cross-validation to yield performance improvements in hyperparameter search subject to a time budget. The investigation is thorough and the paper is clearly written for the most part withholding the suggested amendments to aid clarity above.

&nbsp;

**Potential Impact On The Field Of Automl:**

&nbsp;

The empirical study presented by the authors is practicable and is very likely to be of interest to AutoML practitioners. The generality of the study is also liable to attract a broad readership.

&nbsp;

**Reproducibility:**

&nbsp;

It would be beneficial for users of the package if the modules and functions in e1.py, download.py etc. could be documented e.g. using the PEP 257 convention.

&nbsp;

**Review Confidence:**

4

**Review Rating:**

9

**Review Summary:**

&nbsp;

In summary, the authors conduct a thorough empirical study into the efficacy of early stopping cross-validation for AutoML. I can find no technical flaws in the investigation and the topic is likely to interest a large leadership. As such, I would be very happy to see the paper accepted at the conference.

Update: Now able to edit score, have upgraded!

&nbsp;

**Technical Quality And Correctness:**

&nbsp;

I do not see any issues with the technical correctness of the study although I have not gone as far as to attempt to reproduce the results by running the provided code.

&nbsp;

---

### Official Review · Reviewer_EQKK · 2024-03-28

**Potential Impact On The Field Of Automl Rating:** 3
**Technical Quality And Correctness Rating:** 2
**Clarity Rating:** 3

**Summary Of Contributions:**

This paper proposed to employ early stopping during cross-validation to improve model selection efficiency in automated machine learning systems for tabular data. They propose two early stopping strategies, one more aggressive and the other more forgiving. Through experiments on two algorithms across 36 classification datasets, the study shows that early stopping allows faster convergence in model selection, enables larger exploration coverage, and leads to better overall performance.

**Actions Required To Increase Overall Recommendation:**

Addressing the issue regarding experiments (additional benchmarks) mentioned in the technical quality and correctness section will increase my overall recommendation score.

Addressing the issue regarding theoretical discussion mentioned in the technical quality and correctness section will also increase my overall recommendation score.

**Clarity:**

The writing of the paper is clear for the most part. I note several minor issues below.

- Table 1: Could you clarify what “Datasets Failed” means?
- Line 265: It would be good explain why Aggressive consistently fails to beat No ES.

**Overall Review:**

Positive aspects:
- This paper is well motivated and the proposed early stopping strategies for cross validation are easy to understand
- The proposed algorithms are compared against no ES. The forgiving strategies achieves better coverage, faster convergence, and better overall performance.

Negative aspects:
- Theoretical analysis justifying the proposed algorithms is lacking.
- The baseline algorithm is too simple and no other benchmarks are considered in the experiments.

**Potential Impact On The Field Of Automl:**

This paper could potentially have a significant impact in the field of AutoML because existing literature did not explicitly consider early stopping in the cross validation process for training machine learning models. Hence, it could be of practical use to researchers seeking solutions for early stopping in cross validation.

**Review Confidence:**

4

**Review Rating:**

7

**Review Summary:**

The paper presents good motivation and straightforward early-stopping strategies, but the experimental results can be further improved. In addition, theoretical discussion is missing. Based on these observations, I recommend a weak reject.

**Technical Quality And Correctness:**

This paper has a good motivation and the proposed approach appears reasonable. The two heuristic early-stopping algorithms introduced are easy to understand and leads to good results compared to training models without early-stopping.

However, from my understanding, early stopping has been studied extensively in much existing work, albeit not explicitly targeting early stopping in cross validation (e.g., [1] and [2]). These studies focus on early stopping of the so-called “learning curves” I argue that although such work do not consider cross-validation explicitly, one can treat the mean score during cross-validation  as a learning curve that evolves as more folds are completed. Therefore, considering the extensive study of early stopping in the literature, I believe the baseline comparison (i.e., without early stopping) employed by the authors may be too simplistic. I suggest discussing these existing works that consider early stopping in the context of AutoML in the literature review and compare against them in the experiments.

In addition, although the proposed strategies are straightforward, theoretical discussion is absent. I believe that treating the performance for a specific hyperparameter configuration as a bandit and providing a 'with high probability' proof could help justify the early stopping strategies.

[1] Domhan, Tobias, Jost Tobias Springenberg, and Frank Hutter. "Speeding up automatic hyperparameter optimization of deep neural networks by extrapolation of learning curves." Twenty-fourth international joint conference on artificial intelligence. 2015.

[2] Dai, Zhongxiang, et al. "Bayesian optimization meets Bayesian optimal stopping." ICML 2019.

---

### Official Review · Reviewer_XTUD · 2024-04-02

**Potential Impact On The Field Of Automl Rating:** 3
**Technical Quality And Correctness Rating:** 3
**Clarity:** The paper is well-written and easy to…
**Clarity Rating:** 4

**Summary Of Contributions:**

This paper proposes to investigate early-stopping (ES) methods of cross-validation for AutoML systems. Two heuristics, one being very similar to ROAR and the other being less aggressive/greedy, are extensively tested on the model selection task of classification problems. The paper is, in general, well-written, and the results seem reproducible. The main contributions are: (1) to investigate the effectiveness of early stopping in CV for the random-search tuner and (2) the generalizability of the method. Also, the paper attempts to visualize the differences among w/ and w/o ES methods in the search spaces via dimensionality reduction.

**Actions Required To Increase Overall Recommendation:**

- please clarify why irace, hyperhand, or other tuning methods are not considered in the experiment.
- please explain what we see from Fig. 2 better.
- please provide insights into the non-generalizability of MLP, as shown in Fig. 4.

**Overall Review:**

Pros:

- well-written paper
- crystal clear research questions/hypothesis
- solid results and visualizations
- sufficient details (in the appendix) for reproducing the results

Cons:

- I am not fully convinced why it is nonsensible to also compare to iRace: you can turn off the adaptation of sampling distribution in iRace and simply do a random search. I am very curious about the performance comparison between your approach and iRace.
- I think the authors should also compare this ES approach to Hyperband, which aims to early-stop the configurations in parallel.
- Can the authors provide insights into the non-generalizability of MLP, as shown in Fig. 4?
- I am a bit puzzled by Fig. 2: if we use a random search, then the probability distribution of the sampling configuration should be about the same. What is causing the discrepancy in Fig. 2? random error?

**Potential Impact On The Field Of Automl:**

This paper addresses an important yet sometimes overlooked techique - the early stopping method. IMHO, I think it could be an important contribution if the authors also test the approach in a broader sense - with other AutoML algorithms, e.g., HyperBand. I am aware of the discussion of why iRace is not applicable to this study. But, I am not fully convinced why it is nonsensible to also compare to iRace: you can turn off the adaptation of sampling distribution in iRace and simply do a random search. I am very curious about what the performance comparison would be between your approach and iRace.

**Review Confidence:**

4

**Review Rating:**

8

**Review Summary:**

As I said above, the paper is well-written, and the research questions are super clear, which are properly answered by the experiments. The limitation of the paper is that the experimental comparison is restricted to random search and BO, which narrows down the paper's potential impact. Hence, I recommend an acceptance.

**Technical Quality And Correctness:**

Overall, the technical quality is very good, and I did not see any major mistakes.

---

### Meta-Review · Area_Chair_ez6a · 2024-04-22

**Paper Recommendation:** Accept
**Confidence:** 5

**Metareview:**

This paper proposes to use early stopping during an AutoML tool's cross-validation procedure with the goal to improve efficiency.

The reviewers agree that the paper, while having some debatable detail aspects, is mature enough for publication. All the reviews are on the accept side, and the discussions seem to have resolved some earlier doubts.

I am hence happy to recommend acceptance of the paper.

---

### Decision · Program_Chairs · 2024-04-29

**Decision:**

Accept

**Comment:**

Thank you for submitting your paper. We are happy to tell you that we accept your paper to the main track. See you in Paris.